# Evaluating the effectiveness of care coordination interventions designed and implemented through a participatory action research process: Lessons learned from a quasi-experimental study in public healthcare networks in Latin America

**María-Luisa Vázquez**[1], **Andrea Miranda-Mendizabal**[1,2], **Pamela Eguiguren**[3], **Amparo-Susana Mogollón-Pérez**[4], **Marina Ferreira-de-Medeiros-Mendes**[5], **Julieta López-Vázquez**[6], **Fernando Bertolotto**[7], **Ingrid Vargas**[1]*, for Equity LA II[¶]

1 Health Policy and Health Services Research Group, Health Policy Research Unit, Consortium for Health Care and Social Services of Catalonia, Barcelona, Spain, 2 School of Medicine and Health Sciences, International University of Catalonia (UIC), Sant Cugat del Vallès, Spain, 3 Escuela de Salud Pública Dr. Salvador Allende Gossens, Facultad de Medicina, Universidad de Chile, Santiago de Chile, Chile, 4 Escuela de Medicina y Ciencias de la Salud, Universidad del Rosario, Bogotá, Colombia, 5 Grupo de Estudos de Gestão e Avaliação em Saúde, Instituto de Medicina Integral Prof. Fernando Figueira, Recife, Brasil, 6 Instituto de Salud Pública, Universidad Veracruzana, Veracruz, México, 7 Facultad de Enfermería, Universidad de la República, Montevideo, Uruguay

¶ The complete membership of the research group can be found in the Acknowledgments.
* ivargas@consorci.org

## Abstract

### Background

Despite increasing recommendations for health professionals to participate in intervention design and implementation to effect changes in clinical practice, little is known about this strategy's effectiveness. This study analyses the effectiveness of interventions designed and implemented through participatory action research (PAR) processes in healthcare networks of Brazil, Chile, Colombia, Mexico and Uruguay to improve clinical coordination across care levels, and offers recommendations for future research.

### Methods

The study was quasi-experimental. Two comparable networks, one intervention (IN) and one control (CN), were selected in each country. Baseline (2015) and evaluation (2017) surveys of a sample of primary and secondary care doctors (174 doctors/network/year) were conducted using the COORDENA® questionnaire. Most of the interventions chosen were based on joint meetings, promoting cross-level clinical agreement and communication for patient follow-up. Outcome variables were: a) intermediate: interactional and organizational factors; b) distal: experience of cross-level clinical information coordination, of clinical management coordination and general perception of coordination between levels. Poisson regression models were estimated.

**Data Availability Statement:** Data used in this study are available for download from the repository of the Equity-LA II project at the public link: http://www.equity-la.eu/en/publicaciones.php?t=IS. For further information regarding the database, please contact to the corresponding author (ivargas@consorci.org).

**Funding:** The research leading to these results, the Equity LA II project, received funding from the European Commission Seventh Framework Programme (FP7/2007–2013) under grant agreement number 305197 (https://ec.europa.eu/research/fp7/index_en.cfm). The funder had no role in study design, data collection and analysis, decision to publish, or preparation of the manuscript.

**Competing interests:** The authors have declared that no competing interests exist.

## Results

A statistically significant increase in some of the interactional factors (intermediate outcomes) -knowing each other personally and mutual trust- was observed in Brazil and Chile INs; and in some organizational factors -institutional support- in Colombia and Mexico. Compared to CNs in 2017, INs of Brazil, Chile, Colombia and Mexico showed significant differences in some factors. In distal outcomes, care consistency items improved in Brazil, Colombia and Uruguay INs; and patient follow-up improved in Chile and Mexico. General perception of clinical coordination increased in Brazil, Colombia and Mexico INs. Compared to CNs in 2017, only Brazil showed significant differences.

## Conclusions

Although more research is needed, results show that PAR-based interventions improved some outcomes regarding clinical coordination at network level, with differences between countries. However, a PAR process is, by definition, slow and gradual, and longer implementation periods are needed to achieve greater penetration and quantifiable changes. The participatory and flexible nature of interventions developed through PAR processes poses methodological challenges (such as defining outcomes or allocating individuals to different groups in advance), and requires a comprehensive mixed-methods approach that simultaneously evaluates effectiveness and the implementation process to better understand its outcomes.

## Introduction

Health services fragmentation is considered one of the main obstacles to attaining effective health care outcomes in many healthcare systems around the world, particularly in Latin America [1], where existing evaluations point out deficiencies in clinical information exchange between care levels (i.e. primary care (PC) and secondary care (SC)) [2, 3], difficulties in patient access to secondary care (SC) [4] and disagreement over treatments or referrals [5, 6]. In response to this challenge, a wide range of interventions to improve care integration have been implemented, primarily in high income countries [7–10], but also in regions such as Latin America where average income are lower [1, 3, 11, 12]. These range from a single care coordination mechanism–based on standardization of processes/skills (e.g. clinical guidelines, ongoing medical training) or feedback between professionals (e.g. multidisciplinary teams, joint clinical case conferences or integrated information systems)–to combining various mechanisms in a more comprehensive approach [7].

The limited evidence on the effectiveness of care coordination interventions, mostly from the United States and Europe [8, 9], shows that their effects depend on contextual factors and how they are implemented [8, 7]. In this regard, collaborative and bottom-up approaches such as participatory action research (PAR) are attracting attention within mainstream health services research because they have the potential to bridge the gap between evidence and practice that is usually found in interventions designed in advance and not adapted to the local context [13, 14]. PAR has been defined as a systematic inquiry, with the collaboration of those affected by the issue being studied, for the purposes of education and taking action or effecting social change [14]. Its key characteristics include a cyclical process of planning, action and

evaluation, both flexible and reflexive, as well as a research partnership that encourages participation [13]. Throughout the PAR process, including decision-making, the practitioners can take responsibility for their own learning, research and practice development and view themselves as an essential component in the achievement of quality care. It also lends greater relevance and validity to interventions designed to resolve common practical problems in their working day, creates more interest and positive feedback for the changes being instituted and ensures the sustainability of the effects of the intervention over time [13, 15].

In the field of health, the PAR approach has been most widely used in community-based interventions, for many years in low and middle income countries, and more recently also in high income countries [14, 16]. In health services, PAR studies tend to focus on hospital nursing [13, 17–19], but its application in improving coordination across care levels is rare [20]. It is in community-based research in which the largest number of evaluations of the effectiveness of interventions developed through PAR processes is found, addressing behavioural changes (e.g. immunization rates, cancer screening, among others), or other by-products related to the PAR process such as the emotional support, empowerment or well-being of the participants. However, there is insufficient evidence and too much variation in existing studies to establish the effectiveness of PAR as a method for planning and implementing interventions [16].

Furthermore, there are few evaluations of either care coordination interventions[9] or PAR [16] that simultaneously analyse effectiveness and the implementation process (including the outer and inner context, the content of the intervention, and the process of implementation) [21]. However, many authors [7, 22–24] point out the importance of evaluating both aspects in order to determine how context, content and process influence the effectiveness of integration strategies. They also suggest conducting more quasi-experimental research to develop theory on the effectiveness of care coordination interventions in real life settings [7, 22, 24].

To sum up, despite the growing recognition of the need to implement interventions using a participatory approach in health services, the evaluation of the effectiveness of such interventions remains a challenge.

## A comprehensive conceptual framework for the analysis of care coordination

The study is oriented by a comprehensive conceptual framework on care coordination across care levels, also described in previous publications [25–27], that addresses the different types and dimensions of clinical coordination, the care coordination mechanisms to improve them and influencing factors. Care coordination is defined here as the harmonious connection of the different services needed to provide care to a patient throughout the care continuum in order to achieve a common objective without conflicts [28]. Two interrelated types of clinical care coordination are distinguished: clinical information coordination, or the exchange and use of the patient clinical information needed to coordinate activities between providers, and clinical management coordination, or the provision of care in a sequential and complementary way, which encompasses three dimensions—care coherence, follow-up and accessibility across levels of care. Clinical care coordination can be influenced by two types of factors: a) organizational, such as the existence of certain types of coordination mechanisms across care levels or having enough time to use them; and b) factors related to professionals, such as values and attitudes towards coordinating care and knowing the professionals of the other care level [26, 27]. The wide range of care coordination strategies which can be used, either based on *standardization of processes/skills* (e.g. clinical guidelines, ongoing medical training) or on *feedback between professionals*, (e.g. multidisciplinary teams, joint clinical case conferences or integrated information systems), can have a direct impact on one or several types and dimensions of

clinical care coordination, and/or an indirect impact due to the way they affect the factors that influence coordination. In short, clinical care coordination is a complex phenomenon, and its analysis requires the use of comprehensive instruments from different perspectives [25]. Firstly, from the health services' perspective, the analysis can be conducted based on the experience and opinions of the professionals involved, by means of surveys or qualitative studies, and/or based on a review of records, using health services indicators. Secondly, it can be performed from the patients' perspective, based on how individual patients experience the coordination of services, what is often referred to as continuity of care [29].

## The Equity-LA II project, the PAR process of designing and implementing interventions to improve clinical coordination, and interventions implemented

This study is part of a wider implementation research project (Equity-LA II) [25] that aims to evaluate the effectiveness of interventions, designed and implemented through PAR processes, to improve clinical coordination and continuity of care between care levels in public healthcare networks of Argentina, Brazil, Chile, Colombia, Mexico and Uruguay. Equity-LA II takes a two-fold approach, combining qualitative research methods to evaluate the process of designing and implementing the interventions [21] and their effects on care coordination from the participants' viewpoint [27] with quantitative methods to analyse their effectiveness in terms of care coordination and continuity of care outcomes in the intervention network [30].

The qualitative evaluation, whose results have been published in two previous papers [21, 27], focused on the implementation process and the intervention outcomes. It enabled the research team to determine, firstly, the factors related to the context, the PAR process and the content of the interventions that influenced the implementation of the interventions, and their interaction across time, and secondly, the contribution of the interventions to the improvement of care coordination across levels of care and the influencing factors, from the intervention participants' viewpoint. The results presented in this paper, which focuses on the analysis of the effectiveness of interventions in improving care coordination across the whole healthcare network, complement the qualitative evaluation results.

The participatory action research (PAR) process began with establishing the local steering committee (LSC) in each country to lead the design and implementation processes, comprising managers from the different care levels in the intervention network (IN), in some cases also health professionals [21], and the research team as a facilitator. The baseline results on clinical coordination in the networks were presented and discussed with the IN professionals. These results showed poor performance of all networks studied in the exchange of information and communication for the follow-up of patients between care levels and, to a lesser extent, inappropriate referrals and disagreement over treatments [26]. The factors found to be associated with a better general perception of coordination were related to work, organization and interaction between doctors: being a secondary care (SC) doctor, considering that there is enough time for coordination within consultation hours, job and salary satisfaction, identifying the primary care (PC) doctor as the coordinator of patient care across levels, knowing the doctors of the other care level and trusting in their clinical skills [26].

In the second phase, an inter-level professional platform (PP) was created: a working group that problematized the baseline results and selected care coordination problems and interventions to improve them. Subsequently, the interventions were designed and implemented through participatory processes. Although these processes differed across countries, in terms of duration, number of PAR cycles and levels of participation, three types of participatory processes can be distinguished: 1) two PAR cycles: i) a short initial design by the PP and/or LSC,

implementation and ii) adjustment and implementation, in Colombia, Brazil and Mexico; 2) longer design with greater participation in working groups and LSC with pilot tests (about 129 participants) and implementation, in Chile; 3) open design to be specified with the PP and subsequent implementation, in Argentina and Uruguay [21]. Due to difficulties related to the context (interference of the political cycle) and the research team (turnover of team members), the intervention in Argentina (shared clinical guidelines for hypertension and diabetes) was not implemented or adopted by healthcare professionals [31]. Therefore, no change in coordination between levels of care in the network attributable to the intervention could be expected, hence no data from Argentina could be analysed in this article.

The main problems selected in all countries were related to a lack of communication, agreement over clinical management and exchange of clinical information between PC and SC doctors [31]. Different types of interventions, with implementation stages that varied in duration, were implemented to address the selected problems within the framework of the project: joint meetings of PC and SC doctors to discuss clinical cases and/or for training, either face-to-face (Brazil -7 months, Colombia -16 months, Mexico -6 months) or online (Chile -13 months); implementation of shared clinical guidelines for diabetes (Brazil -21 months); a multi-component strategy to promote the use of the referral and back-referral form (Uruguay-7 months); offline virtual consultations between PC and SC doctors, by email (Brazil-3 months) or through a digital platform (Mexico-14 months); and the design (in cross-level joint meetings) of an induction program to promote a common identity and shared vision (Chile-13 months). The characteristics of the interventions implemented are described in Table 1 and in more detail in previous papers [21].

The research was carried out in six middle-income Latin American countries. Although the study countries have different health system models, as described in a previous paper [26], they are all segmented by population groups according to socioeconomic or employment status [32], with a public subsystem and a private one. The financing of the public sector, which is the focus of this study, is through social security contributions and/or taxes and is mainly intended for the lower income population and/or those without social security. In all six countries, the norms envisage health care organized by levels of complexity, with PC as the entry point and coordinator of patient care and SC care in a supporting role for more complex cases, requiring a referral from PC for access to the specialist [33]. SC includes different types of health care: outpatient secondary care, emergency care and inpatient care. PC and SC doctors in all countries are generally employed on a salary basis, except for SC doctors in Colombia, most of whom are paid on a fee-for-service basis.

The aim of this paper is to evaluate the effectiveness of interventions designed and implemented through participatory action research (PAR) processes in healthcare networks of Brazil, Chile, Colombia, Mexico, and Uruguay to improve clinical coordination across care levels, and to offer recommendations for future research.

## Materials and methods

### Study design and study areas

A quasi-experimental study (a controlled before-and-after design) [34, 35] was carried out in healthcare networks of Brazil, Chile, Colombia, Mexico and Uruguay to evaluate the effectiveness of interventions designed and implemented through participatory action research (PAR) processes in improving clinical coordination between care levels in each country. Baseline and evaluation measurement of care coordination was carried out by means of a survey of primary and secondary care doctors using the COORDENA questionnaire [26].

**Table 1. Characteristics of the interventions and type of PAR process developed for the selection, design and implementation of the interventions.**

| Country | Intervention | Characteristics | Type of participant | N° sessions/ consultations | Duration | PAR process developed |
|---|---|---|---|---|---|---|
| Brazil | Diabetes shared care guidelines | Creation and implementation of diabetes shared care guidelines (focused on essential practices and care pathways) | PC doctors, endocrinologists and other professionals | 9 sessions for creation of shared care guidelines[a] | 21 months | Two PAR cycles: i) short initial design by the LSC and PP, implementation and ii) adjustment and implementation |
| | Joint meetings for discussions of clinical cases in mental health | Discussion of clinical cases (mental health), face-to-face | PC teams and psychiatrists | 11 sessions | 7 months | |
| | Virtual consultation between levels | Asynchronous virtual consultations in mental health via email | PC doctors and psychiatrists | 11 consultations | 3 months | |
| Chile | Joint meetings for discussion of clinical cases | Online clinical conferences (discussion of clinical cases, referral criteria and follow-up) (any condition) | PC and SC doctors and other professionals | 21 sessions[b] | 13 months[b] | Longer design with multiple PAR cycles with pilot tests and implementation and greater participation in working groups and LSC |
| | Induction program for working in network | Cross-level bidirectional visits between PC and SC professionals to promote knowing each other in person, information chart and audio-visual dossier on the network | Professionals of both care levels, but focusing on those starting to work in the healthcare network | 4 cross-level visits[b] | 13 months[b] | |
| Colombia | Joint meetings for discussion of clinical cases and medical training | Discussion of clinical cases and medical training (chronic diseases), face-to-face | PC and SC doctors and other healthcare professionals | 37 sessions | 16 months | Two PAR cycles: i) short initial design by the LSC and PP, implementation and ii) adjustment and implementation |
| Mexico | Joint meeting for medical training | Training sessions based on clinical cases (maternal and perinatal care and chronic diseases) | PC and SC doctors | 5 sessions | 6 months | Two PAR cycles: i) short initial design by the PP and/or LSC, implementation and ii) adjustment and implementation |
| | Virtual communication system between levels | Asynchronous virtual consultations for chronic diseases and maternal and perinatal care via digital platform, and clinical protocols/guidelines repository | PC and SC doctors | 11 consultations; accessed 208 times to look up information | 14 months | |
| Uruguay | Strategy to promote use of referral and reply letter | Standardized format, flowchart and rules of use | PC and SC doctors and other professionals | - | 7 months | Open design to be specified with the PP and subsequent implementation |

PAR—Participatory Action Research; PC—Primary care; SC—Secondary care; PP—professional platform; LSC—local steering committee;

[a] Includes clinical case conferences for creation of shared care guidelines;

[b] Includes pilot tests of the intervention;

More details on the characteristics of each intervention in: http://www.equity-la.eu/es/publicaciones.php?t=AR.

Source: [21, 27].

In each country two comparable healthcare networks (HNs)–an intervention network (IN) and a control network (CN)- were selected according to the following criteria: a) provision of a continuum of services to a defined population including at least primary and secondary care; b) mainly in urban areas of low or medium-low socioeconomic status; and c) willingness to participate and, in the case of the intervention network, to implement designed interventions. Participation was voluntary and there were no refusals to participate from any of the networks contacted. Selected HNs were: Brazil, Districts III and VII in Recife (CN) and the urban area of Caruaru (IN); Chile, southern network (CN) and northern network (IN) of Santiago, encompassing three districts; Colombia, southern district (CN) and southwestern district (IN)

networks of Bogotá; Mexico, municipal networks of Xalapa (CN) and Veracruz (IN); and Uruguay, two networks of the western region, encompassing seven districts.

## Study population and sample

The study population consisted of PC and SC doctors whose daily practice involved contact with doctors of the other care level (i.e. through the patient referral process) and who had been working in the HNs for more than three months. A sample size of 348 doctors per country (174 in each network) and year was estimated to ensure detection of a 15% difference, between HNs and years, in the intermediate and distal outcomes for clinical coordination. It was calculated based on 80% statistical power and a confidence level of 95%. The participation of the professionals in the survey was voluntary. The sample was selected from the list of physicians meeting the criteria and working at the network centres at the time of survey, thus the doctors in the two samples were not necessarily the same. Following the presentation of the study in the network, the interviewers contacted the doctors selected and verified that they met the inclusion criteria. Since the total number of doctors in each network was relatively small, all those who fulfilled the criteria were invited to participate. The percentage of contacted doctors that refused to participate ranged from 2.65% in Colombia to 7.6% in Uruguay in 2015 and from 3.2% in Chile to 11.6% in Mexico in 2017.

## Data collection

Data were collected by means of face-to-face interviews using the COORDENA® questionnaire (www.equity-la.eu), which was adapted and piloted in each country before the baseline and was modified to include specific questions after the intervention study. Detailed information on the design, structure, adaptation and validation of the questionnaire has already been published [12, 26]. The COORDENA questionnaire is divided into several sections. The first (the focus of this paper) includes 13 items to measure clinical care coordination across levels of care experienced by doctors (clinical information and management coordination), and 1 item on the general perception of clinical care coordination. This is followed by a section on doctors' interactional factors. Answers to the items in both sections are collected by a Likert scale (always, often, sometimes, rarely, never). The third and fourth sections of the questionnaire refer to the knowledge and use of care coordination mechanisms. The questionnaire applied in the evaluation survey also included a specific subsection to analyse the knowledge and use of the implemented interventions. The penultimate section refers to organizational and employment factors and job-related attitudes, and the final one to demographic characteristics. The baseline survey was conducted from May to October 2015 (in Uruguay until May 2016) and the evaluation survey (post-intervention) from November 2017 to January 2018.

## Variables of analysis

To analyse the effectiveness of the interventions in the networks, the items from the COORDENA questionnaire measuring intermediate (influencing factors of cross-level clinical coordination) and distal outcomes (experience and general perception of coordination between care levels) were selected. The selection, oriented by the theoretical framework described in the introduction section [25–27], was based on a prior analysis plan, in which the expected outcomes of the interventions in improving clinical care coordination across care levels were defined (hypothesis).

  Intermediate outcomes: interactional factors (knowing the doctors of the other care level, trust in the clinical skills of doctors of the other care level, identification of PC doctors as coordinators of patient care across care levels) and organizational factors (identification of managers

of PC or SC centres as facilitators of clinical coordination). Distal outcomes: experience of cross-level clinical information coordination, experience of cross-level clinical management coordination, and the general perception of coordination between care levels. Clinical information coordination was measured as the exchange and use of clinical information between care levels [29]. Clinical management coordination encompassed care consistency (agreement over the treatments prescribed by doctors of the other care level, contradictions and/or duplications in the treatments prescribed by the other care level, and repetition of tests already performed at the other care level) and the adequacy of patient follow-up between care levels (SC referral to PC for patient follow-up, SC recommendations to PC for patient follow-up and PC doctors' consultations with SC doctors to clear up any doubts about patient follow-up) [29]. Intermediate and distal outcomes variables were dichotomised by merging original answer options into: high (always/often = 1) or low (sometimes/rarely/never = 0).

## Analysis

The unit of analysis is the healthcare network. For each country, descriptive analyses stratified by network and year were performed. Bivariate analyses using the chi-square test were performed to identify any significant differences in the samples between years and networks in each country. P-values <0.05 were considered significant. Poisson regression models with robust variance were estimated to test the hypotheses on the possible influence of the intervention on intermediate and distal outcomes. Prevalence ratios (PR) and respective 95% confidence intervals (CI 95%) were calculated for each country, in order, firstly, to identify differences in the prevalence of the outcome variables post-intervention with the baseline for each (IN and CN) network; and secondly, to identify differences in the prevalence of the outcome variables in the IN and CN both at baseline and post-intervention. Crude and adjusted analyses by sex, age and care level were performed to control for possible changes in outcomes unrelated to the interventions.

## Ethical considerations

In accordance with the current legislation and regulations in each country, the project has been submitted and approved by the corresponding ethical committees in the countries involved: Clinical Research Ethics Committee, CEIC-Parc de Salut Mar, Spain; Institutional Research Board, Institute of Tropical Medicine, Belgium; Research Ethics Committee, School of Medicine and Health Sciences, University of El Rosario, Colombia; Ethics Committee for Research on Humans, Institute of Integrative Medicine Prof. Fernando Figueira, Brazil; Bioethics Committee and Southern Metropolitan Health Service Ethics Committee, Chile; Health Services Research Ethics Committee of Veracruz State, Mexico; Ethics Committee, School of Nursing, University of the Republic of Uruguay.

Conditions of study procedure, risk and benefit evaluation, confidence and privacy, and informed consent were approved by the ethical committees in the participating countries. In addition, confidentiality agreements were signed with all participating institutions. All interviewees participated on a voluntary basis, after signing an informed consent. The right to refuse to participate or withdraw from the survey, anonymity, confidentiality and protection of data were all guaranteed.

## Results

### Sample characteristics

A total of 1,798 doctors participated in the evaluation survey (baseline n = 1,810). In most samples the majority were men (from 50.3% Chile to 68.1% Colombia), except in Brazil and

Uruguay where more than half were women (Table 2). Age distribution was similar in all countries, although Colombia and Chile had a higher percentage of young doctors (23–35 years;>40%) and Mexico of doctors over 50 years of age (>50%). More than half were SC doctors (from 51.8% Chile to 75% Colombia) and had been working in their centres for more than 3 years, especially in Mexico and Uruguay (>80%). In Brazil, Chile and Colombia a significant proportion of doctors (>30%) had been working in their centres for less than one year. More than 60% of the doctors had a permanent contract, except in Chile and Colombia (<35%). Except for Colombia, a significant proportion (around 50%) worked in the private sector, especially in Uruguay (>80%).

Regarding organizational factors, except for Colombia and Mexico where almost all PC doctors reported to have more than 15 minutes per patient, less than 40% of doctors in the other countries reported this, with a notably low percentage in Uruguay (<10%). In contrast, more than half of SC doctors reported to have more than 15 minutes per patient, except for Brazil (<40%) and Uruguay (<30%). Less than half of the doctors considered that the time available for clinical coordination was enough. Only a small minority (<20%) in all countries declared to have plans to change jobs and the majority (over 70%) were satisfied with their jobs. However, less than half in Brazil, Mexico and Uruguay were satisfied with the salary.

In the comparison of the INs evaluation and baseline samples (S1 Table), some differences were detected. In Colombia, there were fewer young doctors (23–35 years old) and more doctors in the older age bracket (36–77 years old) (p<0.05). Regarding work experience in their centres, Brazil had a lower proportion of doctors with more than three years, and higher in the other two categories (p<0.01). Conversely, in Chile fewer doctors had less than one year's experience and there were more in the other two categories (p<0.05).

Around 39% of all doctors in the INs were aware of the interventions, with a particularly high percentage in Chile for the joint clinical meetings via videoconference (63.5%) and notably low ones for the mental health joint meetings in Brazil (12.8%) and the cross-level bidirectional visits carried out by PC and SC teams in Chile (12.4%). Use of/participation in the interventions varied considerably between countries and interventions (around 40% in Brazil and Mexico, for the first intervention, and 3.3% in Mexico for the second intervention) (Table 2).

Due to the large number of results for each country, changes in the intermediate and distal outcomes in the control networks (S2 and S4 Tables), differences between IN and CN at baseline (S3 and S5 Tables), and frequencies of each item of clinical coordination (S6–S8 Tables) are presented as supporting information.

## Changes in the intermediate outcomes: Factors influencing clinical coordination

To facilitate description, country names are used to refer to IN results in the evaluation survey compared to the baseline, unless a different network or period is specified.

An improvement in the interactional factors was observed in Brazil, Chile and Colombia. Knowing the doctors of the other care level personally (PR:1.65; 95%CI 1.03–2.64), trust in their clinical skills (PR:1.19; 95%CI 1.01–1.40) and the identification of PC doctors as coordinators of patient care across care levels (PR:1.19; 95%CI 1.04–1.37) increased in Brazil. The latter also increased in Chile (PR:1.09; 95%CI 0.94–1.27), as did knowing the doctors of the other care level in Colombia (PR:2.48; 95%CI 0.95–6.47), but with no statistical significance (Table 3). Compared to the CN, knowing doctors of the other care level was higher in Brazil (PR:1.72; 95%CI 1.06–2.79) and lower in Colombia (PR:0.49; 95%CI 0.29–0.85); the latter result was similar at baseline. Trust in the clinical skills of doctors of the other care level was

**Table 2. Demographic, employment and organizational characteristics of the sample of the evaluation survey (2017), by country and network.**

| | Brazil | | Chile | | Colombia | | Mexico | | Uruguay | |
|---|---|---|---|---|---|---|---|---|---|---|
| | Intervention network | Control network | Intervention network | Control network | Intervention network | Control network | Intervention network | Control network | Intervention network | Control network |
| | N = 180 | 182 | 178 | 195 | 182 | 180 | 181 | 181 | 176 | 175 |
| | n (%) | n (%) | n (%) | n (%) | n (%) | n (%) | n (%) | n (%) | n (%) | n (%) |
| **Sex** | | | | | | | | | | |
| Male | 101 (56.1) | 54 (29.7) | 98 (55.1) | 92 (50.3) | 124 (68.1) | 115 (63.89) | 114 (62.9) | 104 (57.5) | 72 (40.9) | 74 (42.29) |
| Female | 79 (43.9) | 128 (70.3) | 80 (44.9) | 91 (49.7) | 58 (31.9) | 65 (36.11) | 67 (37.1) | 77 (42.5) | 104 (59.1) | 101 (57.71) |
| **Age** | | | | | | | | | | |
| 23 to 35 years | 68 (37.8) | 44 (21.1) | 77 (43.3) | 90 (49.1) | 78 (42.9)** | 84 (46.7) | 18 (9.9) | 13 (7.2) | 34 (19.3) | 25 (14.3) |
| 36 to 50 years | 78 (43.3) | 70 (38.5) | 59 (33.1) | 53 (29.0) | 61 (33.5)** | 55 (30.6) | 51 (28.2) | 67 (37.0) | 79 (44.9) | 82 (46.8) |
| 51 to 77 years | 33 (18.3) | 68 (37.4) | 41 (23.6) | 40 (21.9) | 43 (23.6)** | 41 (22.8) | 112 (61.9) | 101 (55.8) | 63 (35.8) | 68 (38.9) |
| **Healthcare level** | | | | | | | | | | |
| Primary care | 69 (38.3) | 55 (30.2) | 80 (44.9) | 86 (47.0) | 81 (44.5) | 45 (25.0) | 87 (48.1) | 61 (33.7) | 50 (28.4) | 66 (37.7) |
| Secondary care | 110 (61.1) | 127 (69.8) | 98 (56.1) | 97 (53.0) | 101 (55.5) | 135 (75.0) | 94 (51.9) | 120 (66.3) | 126 (71.6) | 109 (62.3) |
| **Time working at the centre** | | | | | | | | | | |
| ≤ 1 year | 68 (37.8)*** | 37 (20.3) | 45 (25.3)** | 53 (29.0) | 62 (34.1) | 58 (32.2) | 14 (7.7) | 18 (9.9) | 17 (9.7) | 9 (5.1) |
| > 1 year to 3 years | 39 (21.7)*** | 32 (17.6) | 44 (24.7)** | 38 (20.8) | 46 (25.3) | 32 (17.8) | 16 (8.9) | 17 (9.4) | 19 (10.8) | 27 (15.4) |
| > 3 years | 73 (40.5)*** | 113 (62.1) | 89 (50.0)** | 92 (50.2) | 74 (40.6) | 90 (50.0) | 151 (83.4) | 146 (80.7) | 140 (79.5) | 139 (79.5) |
| **Type of contract** | | | | | | | | | | |
| Permanent | 68 (37.8)*** | 141 (77.5) | 50 (28.1) | 62 (33.9)** | 47 (25.8) | 34 (18.9) | 158 (87.3) | 165 (91.2)*** | 129 (73.3) | 125 (71.4) |
| Temporary | 112 (62.2)*** | 41 (22.5) | 128 (71.9) | 118 (64.5)** | 135 (74.2) | 146 (81.1) | 23 (12.7) | 15 (8.3)*** | 47 (26.7) | 49 (28.0) |
| **Contracted hours per week** | | | | | | | | | | |
| ≤ 20 hours | 69 (38.3)** | 97 (53.3) | 15 (8.4) | 9 (4.9)** | 14 (7.7) | 8 (4.4) | 0 (0.0) | 1 (0.6) | 87 (49.7) | 93 (54.4) |
| 20 to 40 hours | 106 (58.9)** | 80 (44.0) | 64 (36.0) | 80 (43.7)** | 59 (32.4) | 65 (36.1) | 179 (98.9) | 174 (96.1) | 68 (38.9) | 59 (34.5) |
| > 40 hours | 5 (2.8)** | 5 (2.7) | 99 (55.6) | 94 (51.4)** | 109 (59.9) | 107 (59.45) | 2 (1.1) | 6 (3.3) | 20 (11.4) | 19 (11.1) |
| **Working in private sector** [a] | 98 (54.4) | 116 (63.7) | 80 (48.2) | 88 (45.1)*** | 64 (35.2) | 63 (35.0) | 95 (52.5) | 81 (44.5) | 154 (87.5) | 159 (90.9) |
| **Time per patient** * | | | | | | | | | | |
| *Primary care* | | | | | | | | | | |
| 15 minutes or less | 43 (62.3) | 16 (29.1)** | 55 (76.4) | 61 (64.9) | 0 (0.0)** | 1 (2.2) | 5 (5.7) | 11 (18.0) | 48 (96.0)** | 57 (86.4) |
| More than 15 minutes | 26 (37.7) | 39 (70.9)** | 17 (21.3) | 32 (34.0) | 81 (100.0)** | 44 (97.8) | 82 (94.3) | 50 (82.0) | 2 (4.0)** | 6 (9.1) |
| *Secondary care* | | | | | | | | | | |
| 15 minutes or less | 88 (80.0) | 84 (66.1) | 40 (42.5) | 57 (56.4) | 28 (27.7) | 34 (25.2) | 17 (18.1) | 23 (19.2) | 98 (77.8) | 68 (62.4) |
| More than 15 minutes | 22 (20.0) | 43 (33.9) | 53 (56.4) | 44 (43.6) | 73 (72.3) | 101 (74.8) | 77 (81.9) | 74 (61.7) | 13 (10.3) | 26 (23.8) |
| **Enough time during consultation for clinical coordination** [b] | 80 (44.4) | 76 (41.8) | 28 (15.7) | 23 (12.6) | 32 (17.6) | 52 (28.9) | 53 (29.3)** | 62 (34.2) | 61 (34.7) | 71 (40.6) |
| **Knowledge of the intervention** | | | | | | | | | | |

*(Continued)*

**Table 2.** (Continued)

| | Brazil | | Chile | | Colombia | | Mexico | | Uruguay | |
|---|---|---|---|---|---|---|---|---|---|---|
| | Intervention network | Control network | Intervention network | Control network | Intervention network | Control network | Intervention network | Control network | Intervention network | Control network |
| | N = 180 | 182 | 178 | 195 | 182 | 180 | 181 | 181 | 176 | 175 |
| | n (%) | n (%) | n (%) | n (%) | n (%) | n (%) | n (%) | n (%) | n (%) | n (%) |
| Intervention 1 | 77 (42.8) | NA | 113 (63.5) | NA | 77 (42.3) | NA | 82 (45.3) | NA | 49 (27.8) | NA |
| Intervention 2 | 23 (12.8) | NA | 22 (12.4) | NA | NA | NA | 66 (36.5) | NA | NA | NA |
| **Use of/participation in the intervention** [b] | | | | | | | | | | |
| Intervention 1 | 72 (40.0) | NA | 37 (20.8) | NA | 64 (35.2) | NA | 76 (42.0) | NA | 36 (20.4) | NA |
| Intervention 2 | 22 (12.2) | NA | 10 (5.6) | NA | NA | NA | 6 (3.3) | NA | NA | NA |

\* Prevalence calculated using the total for the corresponding healthcare level as denominator.

[a] Affirmative answers are shown.

[b] Always + often categories.

\*\* Differences with respect to baseline values of the network (2015) p<0.05.

\*\*\* Differences with respect to baseline values of the network (2015) p<0.01.

**Intervention 1**: *Brazil* shared clinical guidelines for diabetes, *Chile* on-line joint clinical meetings, *Colombia* joint clinical meetings, *Mexico* joint clinical meetings.

**Intervention 2**: *Brazil* mental health joint meetings, *Chile* cross-level bidirectional visits between PC and SC teams, *Mexico* offline virtual consultations between PC and SC doctors.

**NA** Not applicable.

lower in Colombia (PR:0.83; 95%CI 0.71–0.97) and identification of PC doctors as the coordinators of patient care was higher in Brazil (PR: 1.20; 95%CI 1.04–1.40) and Chile (PR:1.17; 95%CI 1.02–1.34) (Table 4).

Organizational factors showed positive changes in Brazil, Colombia and Mexico, without statistical significance in Brazil, after adjustment. There was an increase in those that found the SC centre managers to be facilitators of clinical coordination in Colombia (PR:1.54; 95%CI 1.01–2.35), and in Mexico (PR:1.80; 95%CI 1.28–2.53), where there was also an increase in the same factor with regard to PC centre managers (PR:1.92; 95%CI 1.32–2.80) (Table 3).

**Table 3. Changes in the influencing factors of cross-level clinical coordination (intermediate outcomes) between 2015 and 2017 in the intervention networks of each country.**

| | Brazil | Chile | Colombia | Mexico | Uruguay |
|---|---|---|---|---|---|
| | IN 2015/2017 | IN 2015/2017 | IN 2015/2017 | IN 2015/2017 | IN 2015/2017 |
| | PR (IC 95%)* | PR (IC 95%)* | PR (IC 95%)* | PR (IC 95%)* | PR (IC 95%)* |
| *Interactional factors between professionals* | | | | | |
| Knowing the doctors of the other care level personally | **1.65 (1.03–2.64)** | 0.81 (0.45–1.45) | 2.48 (0.95–6.47) | 1.15 (0.68–1.94) | 0.92 (0.80–1.04) |
| Trusting in clinical skills of doctors of the other care level | **1.19 (1.01–1.40)** | 1.02 (0.88–1.19) | 0.96 (0.82–1.12) | 1.03 (0.86–1.24) | 0.97 (0.89–1.07) |
| Identification of PC doctors as coordinators of patient care across care levels | **1.19 (1.04–1.37)** | 1.09 (0.94–1.27) | 0.87 (0.72–1.04) | 1.06 (0.90–1.24) | 1.18 (0.98–1.42) |
| *Organizational factors* | | | | | |
| PC centre managers facilitate clinical coordination between care levels | 1.38 (0.97–1.95) | 1.03 (0.68–1.56) | 1.32 (0.86–2.03) | **1.92 (1.32–2.80)** | 1.06 (0.76–1.48) |
| SC centre managers facilitate clinical coordination between care levels | 1.19 (0.85–1.66) | 0.88 (0.53–1.46) | **1.54 (1.01–2.35)** | **1.80 (1.28–2.53)** | 1.07 (0.80–1.43) |

\* Adjusted for: sex, age, healthcare level.

IN: intervention network. PR: prevalence ratio. PC: primary care. SC: secondary care.

**Table 4. Differences in influencing factors of cross-level clinical coordination (intermediate outcomes) in 2017 in the intervention and control networks of each country.**

| | Brazil | Chile | Colombia | Mexico | Uruguay |
|---|---|---|---|---|---|
| | IN vs. CN 2017 | IN vs. CN 2017 | IN vs. CN 2017 | IN vs. CN 2017 | IN vs. CN 2017 |
| | PR (IC 95%) | PR (IC 95%) | PR (IC 95%) | PR (IC 95%) | PR (IC 95%) |
| *Interactional factors between professionals* | | | | | |
| Knowing the doctors of the other care level personally | **1.72 (1.06–2.79)** | 1.77 (0.88–3.58) | **0.49 (0.29–0.85)** | 1.26 (0.80–1.97) | 1.05 (0.91–1.21) |
| Trusting in clinical skills of doctors of the other care level | 1.08 (0.92–1.27) | 1.00 (0.88–1.15) | **0.83 (0.71–0.97)** | 1.03 (0.87.1.22) | 1.05 (0.95–1.15) |
| Identification of PC doctors as coordinators of patient care across care levels | **1.20 (1.04–1.40)** | **1.17 (1.02–1.34)** | 0.88 (0.72–1.07) | 1.13 (0.97–1.32) | 1.07 (0.91–1.26) |
| *Organizational factors* | | | | | |
| PC centre managers facilitate the clinical coordination between care levels | 1.44 (0.97–2.14) | 0.74 (0.52–1.06) | 0.78 (0.55–1.10) | 0.95 (0.70–1.29) | 0.76 (0.60–1.01) |
| SC centre managers facilitate the clinical coordination between care levels | 1.43 (0.97–2.10) | 0.77 (0.48–1.24) | 0.85 (0.59–1.20) | 0.80 (0.60–1.06) | 0.91 (0.69–1.20) |

\* Adjusted for: sex, age, healthcare level. IN: intervention network. CN: control network. PR: prevalence ratio. PC: primary care. SC: secondary care.

Compared to the CN, the proportion of doctors that found the PC centre managers to be facilitators of coordination was similar in Mexico, showing improvement over the baseline where it was lower (Table 4).

## Changes in the experience of information and clinical management coordination between care levels

Exchange of information between care levels showed improvements in Brazil, Chile and Mexico. However, these results were non-significant (Table 5). Compared to the CNs, exchange of information was higher in Colombia (PR:1.60; 95%CI 1.25–1.99) and in Mexico without statistical significance (Table 6).

With regard to clinical management coordination, increases in some items were observed in all countries. Consistency of care improved in Brazil, Colombia and Uruguay, specifically in agreement over treatments prescribed by the other care level: Brazil (PR:1.24; 95%CI 1.01–1.52), Colombia (PR:1.20; 95%CI 1.04–1.62) and Uruguay (PR:1.33; 95%CI 1.07–1.64).

**Table 5. Changes in the experience of cross-level coordination of information and clinical management of care (distal outcomes) between 2015 and 2017 in the intervention networks of each country.**

| | Brazil | Chile | Colombia | Mexico | Uruguay |
|---|---|---|---|---|---|
| | IN 2015/2017 | IN 2015/2017 | IN 2015/2017 | IN 2015/2017 | IN 2015/2017 |
| | PR (IC 95%) | PR (IC 95%) | PR (IC 95%) | PR (IC 95%) | PR (IC 95%) |
| *Coordination of information* | | | | | |
| Exchange of information between care levels | 1.50 (0.99–2.27) | 1.34 (0.76–2.36) | 0.88 (0.68–1.15) | 1.37 (0.87–2.15) | 0.97 (0.77–1.22) |
| *Consistency of care across care levels* | | | | | |
| Agreement over the treatments prescribed by the other care level | **1.24 (1.01–1.52)** | 1.13 (0.91–1.40) | **1.20 (1.04–1.62)** | 1.09 (0.87–1.37) | **1.33 (1.07–1.64)** |
| Contradictions and/or duplications in the treatments prescribed by different care levels | 0.76 (0.52–1.11) | 1.19 (0.77–1.85) | 0.72 (0.49–1.06) | 1.04 (0.67–1.61) | 0.76 (0.43.1.34) |
| Repetition of tests that were already performed at the other care level | 0.64 (0.39–1.04) | 1.09 (0.77–1.55) | 0.77 (0.56–1.06) | 1.36 (0.96–1.91) | 0.72–0.39–1.33) |
| PC refers the patient to SC when necessary | 1.02 (0.91.1.14) | 0.87 (0.74–1.02) | 0.96 (0.87–1.06) | **1.19 (1.06–1.34)** | 1.02 (0.92–1.13) |
| *Patient follow-up between care levels* | | | | | |
| SC doctors make recommendations to PC doctors for patient follow-up | 1.27 (0.89–1.81) | **1.32 (1.04–1.66)** | 1.03 (0.77–1.39) | **1.28 (1.03–1.60)** | 0.92 (0.74–1.14) |
| PC doctors consult SC doctors with any queries about patient follow-up | 1.47 (0.89–2.44) | 1.22 (0.72–2.07) | 0.98 (0.76–1.27) | **2.10 (1.06–3.91)** | 0.98 (0.82–1.17) |
| SC refers patients to PC for follow-up | 1.00 (0.85–1.18) | 1.13 (0.94–1.34) | 0.10 (0.05–0.18) | 1.11 (0.92–1.34) | 1.02 (0.82–1.28) |

\* Adjusted for: sex, age, healthcare level. IN: intervention network. PR: prevalence ratio. PC: primary care. SC: secondary care.

**Table 6. Differences in experience of cross-level coordination of information and clinical management of care (distal outcomes) in 2017 in the intervention and control networks, by country.**

| | Brazil | Chile | Colombia | Mexico | Uruguay |
|---|---|---|---|---|---|
| | IN vs. CN 2017 | IN vs. CN 2017 | IN vs. CN 2017 | IN vs. CN 2017 | IN vs. CN 2017 |
| | PR (IC 95%) | PR (IC 95%) | PR (IC 95%) | PR (IC 95%) | PR (IC 95%) |
| *Coordination of information* | | | | | |
| Exchange of information between care levels | 0.90 (0.60–1.34) | 0.72 (0.40–1.29) | **1.60 (1.25–1.99)** | 1.09 (0.73–1.63) | 0.91 (0.70–1.18) |
| *Consistency of care across care levels* | | | | | |
| Agreement over the treatments prescribed by the other care level | 1.21 (0.98–1.49) | 0.95 (0.83–1.11) | 0.86 (0.72–1.03) | **1.24 (1.01–1.54)** | 1.13 (0.94–1.36) |
| Contradictions and/or duplications in the treatments prescribed by different care levels | 1.06 (0.67–1.66) | 1.04 (0.69–1.55) | 1.05 (0.67–1.63) | 0.95 (0.62–1.46) | 1.00 (0.56–1.81) |
| Repetition of tests that were already performed at the other care level | 0.66 (0.40–1.09) | **0.73 (0.55–0.98)** | **1.46 (1.03–2.08)** | 0.93 (0.67–1.28) | 0.67 (0.36–1.23) |
| PC refers the patient to SC when necessary | 0.94 (0.84–1.04) | **0.84 (0.74–0.96)** | 0.95 (0.85–1.05) | **0.87 (0.77–0.98)** | 1.03 (0.92–1.14) |
| *Patient follow-up between care levels* | | | | | |
| SC doctors make recommendations to PC doctors for patient follow-up | 0.83 (0.61–1.12) | 1.20 (0.97–1.49) | **0.79 (0.63–0.99)** | 1.01 (0.83–1.22) | 0.96 (0.77–1.20) |
| PC doctors consult SC doctors with any queries about patient follow-up | 0.76 (0.50–1.15) | 1.33 (0.80–2.23) | **0.75 (0.61–0.93)** | 0.95 (0.56–1.60) | 0.90 (0.76–1.07) |
| SC refers patients to PC for follow-up | 1.03 (0.87–1.23) | 1.08 (0.92–1.26) | **0.72 (0.53–0.99)** | 1.11 (0.93–1.31) | 1.10 (0.88–1.38) |

* Adjusted for: sex, age, healthcare level. IN: intervention network. CN: control network. PR: prevalence ratio. PC: primary care. SC: secondary care.

Likewise, in Mexico there was an improvement in the adequacy of PC referrals (PR:1.19; 95% CI 1.06–1.34) (Table 5). Compared to the CN, agreement over treatments was higher in Mexico (PR: 1.24; 95%CI 1.01–1.54). Duplication of tests was lower in Chile (PR: 0.73; 95%CI 0.55–0.98) and higher in Colombia (PR:1.46; 95%CI 1.03–2.08), and the adequacy of PC referrals to SC was lower in Chile (PR:0.84; 95%CI 0.74–0.96) and Mexico (PR:0.87; 95%CI 0.77–0.98) (Table 6).

Chile and Mexico showed improvement in the adequacy of patient follow-up between care levels: the recommendations of SC doctors to PC doctors for patient follow-up increased (Chile PR:1.32; 95%CI 1.04–1.66; Mexico PR:1.28; 95%CI 1.03–1.60). In Mexico, PC doctors' consultations with SC doctors to clear up any doubts on patient follow-up (PR:2.04; 95%CI 1.06–3.91) also increased (Table 5). Compared to the CN in Colombia, while SC doctors' recommendations to PC doctors (PR:0.79; 95%CI 0.63–0.99) and PC consultations with SC (PR: 0.75; 95%CI 0.61–0.93) were lower, they were similar at baseline. However, SC referrals to PC were also lower (PR:0.72; 95%CI 0.53–0.99), unlike the baseline, where there were no differences. In Mexico, SC doctors' recommendations to PC doctors and adequacy of SC referrals to PC increased without statistical significance (Table 6).

## Changes in the general perception of coordination between care levels

Increased general perception of coordination between care levels was observed in Brazil (PR:2.03; 95%CI 1.16–3.54), Colombia (PR:1.46; 95%CI 1.06–2.01) and Mexico (PR:2.46; 95% CI1.42–4.25). Compared to the CN, it was higher in Brazil (PR:2.04; 95%CI 1.10–3.75), as it was at baseline. In Colombia, it was lower (PR: 0.66; 95%CI 0.52–0.84), while the difference was non-significant at baseline. In Mexico, it was lower like at baseline, but it did lose statistical significance (PR:0.97; 95%CI 0.65–1.43) (Fig 1).

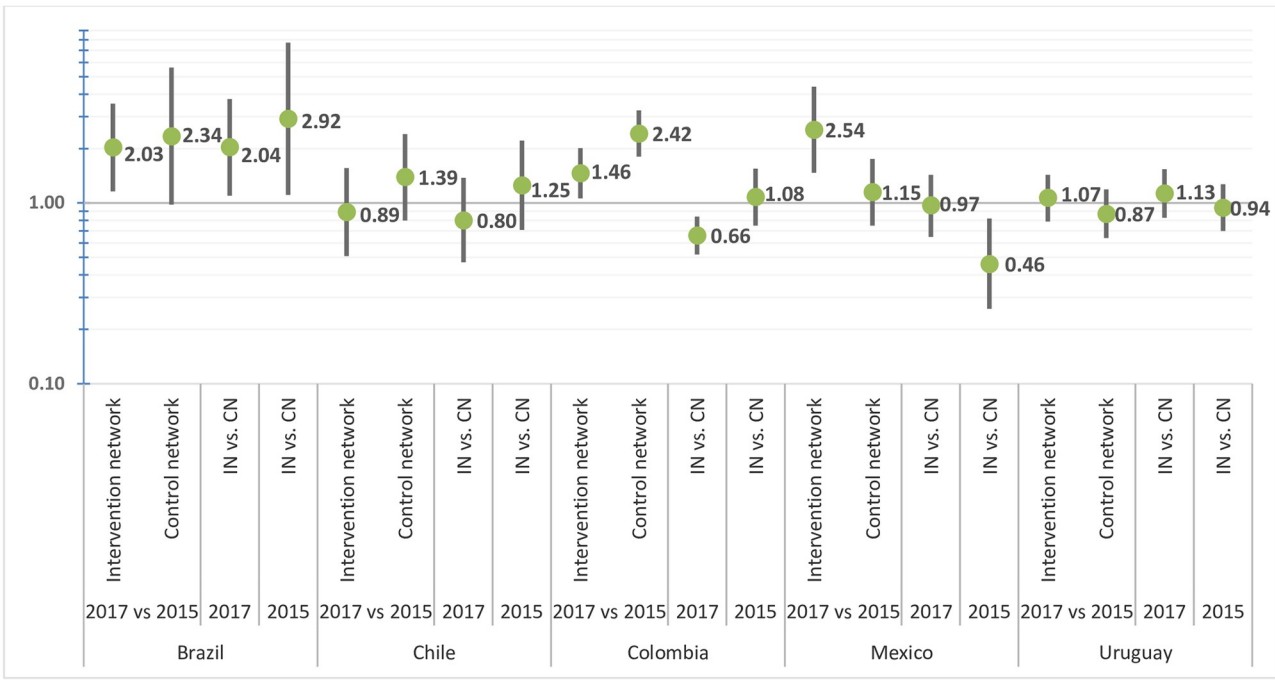

**Fig 1. Changes in general perception of care coordination between 2015 and 2017 in the intervention and control networks, by country.** Results from the Poisson regression models with robust variance adjusted for sex, age and healthcare level. **IN**: intervention network. **CN**: control network.

## Discussion

This study addresses a challenging and little explored area in the field of health services in general and coordination between care levels in particular: evaluating the effectiveness of interventions designed and implemented through participatory action research. Various different approaches and disciplines–implementation research, quality models, transfer of scientific evidence [36–38]–indicate the need for a more collaborative and participatory approach for tailoring interventions and achieving changes in clinical practice and in organizations [39, 40]. However, little is known regarding the effectiveness of participatory interventions in health services. The few existing evaluations have focused on analysing the participatory process [13, 17–19]. Another novel approach is the use of multiple outcome measures of the experience and perceptions of doctors to cover the different aspects of coordination. Some reviews of interventions to improve care coordination [7] highlight the importance of complementing any effectiveness analysis, which generally focuses on distal outcomes such as the health of a given population, with these types of self-reported primary outcomes. This gives a fuller picture of the effectiveness of a care coordination intervention, and helps to determine to what extent the improvement seen in distal outcomes can be attributed to improvements in coordination.

Although more research is needed on the effectiveness of PAR-based interventions on clinical coordination and the most appropriate way to analyse them, this study provides evidence on their impact in improving some items of clinical coordination across care levels (with differences between countries), highlights areas for improvement and also offers lessons and recommendations for future research.

### Improvements in some aspects of clinical coordination relevant to the health professionals in the study networks

In most countries, similar interventions–cross-level joint meetings–were implemented to discuss clinical cases, provide training and establish shared protocols. It was also joint meetings, held by the different participating entities (LSC, PP and working groups), that formed the basis of the participatory process of selection, design and monitoring of the implementation with the stakeholders. Although there were differences between countries, a statistically significant increase was observed in all the INs in some of the items on experience of clinical management coordination related to care consistency and/or coordination of patient follow-up. Furthermore, an improvement was seen in certain interactional factors between doctors in Brazil and Chile, and organizational factors in Colombia and Mexico, as well as in the general perception of cross-level coordination in the networks of Brazil, Colombia and Mexico. Of the 14 quantitative outcomes evaluated, the intervention was associated with 6 statistically significant changes in Mexico, 5 in Brazil, 3 in Colombia and 1 in Chile and Uruguay (S9 Table). In the post-intervention comparison with the control networks, positive results were seen in the INs in some of the items on experience of clinical management coordination and interactional factors in Brazil (2 items), Chile (2 items) and Mexico (1 item), as well as in the general perception of coordination in Brazil and information exchange in Colombia. Some more negative results were also observed in comparison to the CN, mainly in Colombia, in items related to interactional factors, coordination of patient follow-up and perception of coordination (6 items), but with no changes with respect to those observed at baseline.

Various aspects should be highlighted with regard to the improvements achieved: firstly, their consistency with the desired aims of the interventions, i.e. improvement of clinical agreement between PC and SC doctors and communication for patient follow-up; secondly,

their relevance, on referring to problems selected by the health professionals themselves to improve their daily practice [13, 15]; and thirdly, their reach, on being improvements that are not limited to the intervention participants, but rather apply to the whole network of health services. All this represents an achievement, considering the limited time frame of this evaluation.

One of the most important improvements observed was the increase in *agreement over the treatments prescribed* between care levels in all the countries, although it was only significant in Brazil, Colombia and Uruguay, which may be due to the type of intervention implemented (based on mutual adjustment). The results of the qualitative evaluation highlight that the joint clinical meetings, based on doctors reflecting on their own praxis, specifically promoted agreement on patients' treatments and allowed them to adapt or establish joint care protocols taking the local context into account (e.g. restrictions on resources) [27]. In Brazil, the discussion on the role of each care level when drawing up the shared clinical guidelines for diabetes, and their implementation, accompanied by the training of professionals to promote their use, also contributed to promoting clinical agreement. Likewise, the *increase in cross-level coordination of patient follow-up and referrals* observed in Chile and Mexico was identified as a direct consequence of the joint meetings, as well as the increased use of other communication mechanisms (e.g. WhatsApp) that the meetings fostered. Lastly, in all countries except Colombia, a slight *improvement in information exchange*, although not significant, was also perceived by users [30].

The baseline results clearly threw into focus the magnitude of the coordination problems and their relation to interactional factors between doctors of different care levels (barely knowing each other and mutual distrust) [26]. *Improvement in interactional factors* was observed in Brazil, and in Chile when compared to the CN, which was consistent with the results of the qualitative evaluation [21]. The joint clinical meetings created a meeting space that encouraged doctors to get to know each other in person. This, together with the reflexive methods used, helped to improve relationships between care levels, and fostered their willingness to collaborate and motivation to participate [21, 27]. Until now, improvements in personal relationships between professionals have mostly been reported in evaluations of interventions implemented within the same level of care (acute care settings), such as multidisciplinary teams [7]. The results of this study show that interventions based on collaborative work between professionals of different levels achieve similar effects.

The qualitative study [21, 27] also shows the influence of the participatory process in the adoption of interventions, an aspect which may have contributed to their effectiveness, firstly, by generating greater awareness of the care coordination problem and commitment from participants, and secondly, by improving interactional factors, in keeping with previous studies [41, 42].

On a final note, the worse results in Colombia compared to the CN (which were similar to baseline) may be attributable to changes in the municipality that affected the two networks in an uneven way, as the analysis of contextual changes indicates [43]. During this period, the Bogotá Health Department, following national policy changes [44, 45], reorganized its health services network, introducing a number of strategies and clinical care coordination mechanisms such as electronic medical records (EMR), care paths and the co-location of certain specialities in primary care centres and joint clinics [46]. At the time of study, many of these mechanisms had already been established in the control network, whereas the intervention network encountered several organizational problems at the preliminary introduction stage, which led some health professionals to reject them [46]. The effects of the intervention may therefore have been obscured in part by these contextual changes and their favourable effects on care coordination in the control network.

## Adequate time frame and institutional support for implementation: Key to detecting and consolidating changes in participatory interventions

Although the effectiveness study shows impacts on some clinical coordination items across the whole network, the results of the qualitative study were more conclusive on the changes attributed to the implementation of joint meetings [27], especially in Chile and Colombia. This difference, also reported in a previous study analysing effectiveness at area level and the implementation process [47], may be related to a limited penetration of the interventions, which was lower still in these two countries: 20.8% of those surveyed had participated frequently in the joint meetings in Chile, and 35.2% in Colombia (and they were all PC doctors) [21].

Among the reasons for this limited penetration we find, firstly, a short implementation period at the time of evaluation (low intervention "dose"). Thus the time factor is key to detecting changes in interventions based on cross-level collaboration because of their high level of complexity [48]. This is yet more relevant in the case of bottom-up interventions, in which participation tends to be low at first and gradually increases as the participatory process is consolidated [13]. PAR studies in the field of health care are in agreement on a minimum time frame of 3 to 5 years for the implementation of interventions in order to detect significant changes [13, 49]. It is therefore unsurprising that fewer changes were registered at network level in Chile, where the PAR process was more participatory, progressive and sustainable, but the institutionalization of the intervention had only just been achieved at the time of evaluation [27]. The time frame for the design process (several cycles of action-reflection and pilot tests) and widespread dissemination (63.5% of doctors knew of the intervention) was longer (10 months), however the implementation process had only been underway for 3 months at the time of evaluation [31].

Secondly, the low penetration of the joint meetings, especially among SC doctors in Colombia, may be related to the limited institutional support available [21], which led to low participation in the joint meetings. In the case of Colombia, managers' support for the participation of SC doctors in the meetings was greatly hindered by incentives put in place for providers to maximize their healthcare activity in the interests of financial sustainability in a market model, and the limited number of full-time SC doctors working in the network. The results from the Brazilian network confirm the importance not only of an adequate implementation time frame to observe changes at network level (21 months for shared care guidelines), but also of institutional support. This factor is also related to the changes observed in Mexico. In both Brazil and Mexico, there was firm institutional support providing the resources required (professionals' time and materials), as the interventions were aligned with or gave continuity to local party policy [50, 51].

The results also show few quantifiable changes in clinical coordination in the Uruguay IN, which can be attributed to a low uptake of the referral and reply letter implemented (only 27.8% of doctors knew of the intervention, of which 20.4% used it frequently). The qualitative evaluation [27] indicates that the reasons for its low usage, apart from the short implementation time frame (7 months), were the convergence of contextual barriers (e.g. inconsistent institutional support due to changes in government) and deficiencies in the content of the intervention and the PAR process (e.g. limited participation of doctors in the process of selection, design and dissemination). These latter factors may also be related to the low uptake (and thus low network penetration) of other interventions such as joint meetings in mental health in Brazil and offline virtual consultations in Mexico.

## Limitations of the study and recommendations for future evaluations of the effectiveness of interventions based on PAR processes in the health services

Quasi-experimental studies (controlled before-and-after designs) are increasingly recommended by implementation science because they permit the evaluation of interventions under real-life conditions or in routine practice [8, 52, 53], unlike randomized studies. However, certain methodological difficulties were encountered during the course of this study related to the characteristics of the PAR approach. Similar issues have been reported elsewhere [34, 54–56] and are common in participatory action research.

Firstly, the open and flexible nature of the PAR process, in which both problems and interventions are defined progressively in collaboration with the practitioners through iterative cycles of planning, action and evaluation, made it difficult to define in advance the outcome variables related to the specific objectives of the interventions, and to measure them before and afterwards, as previous studies have pointed out [13]. The use of routine indicators related to the expected outcomes could help to resolve this issue, but as these were not available in our case, broader outcomes were used as an alternative. However, the likelihood of detecting changes in broader outcomes is reduced [55].

Secondly, since participation in the PAR process was voluntary and incremental, individuals gradually joined in voluntarily as feelings of empowerment and motivation increased, so it was not possible to allocate clusters of individuals within the organization to different intervention groups. Thus the whole organization at which the intervention was aimed (i.e. the network) had to become the unit of allocation. However, detecting changes at network/organization level poses challenges that have been extensively highlighted in fields such as community-based evaluation [54, 55].

Thirdly, the flexibility of design of the PAR interventions, which allowed the interventions and their components to be adjusted throughout the process [21], also hindered determination of the effectiveness of each component.

Lastly, the PAR process generates increased awareness of the problem, in this case clinical coordination, which might make doctors view it more critically and cause them to rate some aspects more negatively during the evaluation survey in the intervention area. This may be the case in particular when the analysis of the problems is carried out extensively, like in Chile and Colombia, where around 200 healthcare professionals in the intervention network participated in the dissemination and problematization of the results of the baseline study [31], but only a small group of doctors participated in the intervention. This should also be taken into account for the interpretation of the results.

To these difficulties we must add the problems inherent to quasi-experimental designs that hamper the internal validity and generalization of results [34]. Contextual changes (or secular trends), in particular, can make it difficult to attribute observed changes to the intervention, or conversely, mask possible demonstrable effects of the intervention. In order to strengthen the interpretation of causality between the results and the implemented interventions, this study included a control network/organization of similar characteristics, and adjusted analyses for possible confounding variables were conducted [56]. Nevertheless, the results must be viewed with caution due to the difficulties involved in identifying a comparable network and randomizing network selection, given the prevailing influence of political criteria, feasibility or willingness to implement the intervention (especially one that calls for a time-consuming PAR process). Furthermore, it is impossible to prevent the control network from introducing changes during the process, or to avoid cross-contamination if there is a close relationship

between the networks. In summary, analysing the effectiveness of PAR-based interventions involves several methodological and interpretative challenges.

Based on this experience, some general recommendations can be made for future evaluations. Firstly, it is important to adopt a comprehensive mixed-methods approach that simultaneously evaluates effectiveness and the implementation process to identify and understand the context and process-related factors of success or failure for the intervention, the mechanisms by which the intervention has contributed to the associated results [23, 24], and other outcomes not detected by the quantitative large-scale evaluation [55]. Secondly, in the implementation planning and evaluation of interventions based on PAR processes it is essential to consider the significant amount of time (and therefore institutional support) required for these types of interventions. This includes time to negotiate access, understand the context, establish group processes and relationships [13], and to implement and complete a well-developed intervention and rigorous evaluation [57]. A longer implementation time frame will allow for a higher level of penetration of the interventions in the networks and hence deliver measurable impacts [16]. Any evaluation of the effectiveness of interventions based on PAR processes must therefore envisage measurement of changes in the long term, as well as taking into account the long-term research funding mechanisms needed to make it feasible [57].

## Conclusions

This study contributes to filling the gap in knowledge on the effectiveness of PAR-based interventions in health services to improve clinical coordination between care levels. The results, which reflect certain differences between countries, show the impact of interventions at network level on the improvement of some items of coordination considered relevant by health professionals in the study networks, such as agreement over treatments, coordination of patient follow-up and interactional factors. However, the PAR process is slow and gradual, and in order to detect and consolidate the network-level changes identified by the professionals that participated in interventions such as the joint meetings, more implementation time and institutional support is needed, and thus it requires a long-term evaluation with its funding. Finally, this study indicates that the rigorous evaluation of the effectiveness of PAR-based interventions in health services poses considerable methodological and interpretative challenges related to the characteristics of the PAR process. It is therefore important to adopt a comprehensive mixed-methods approach that also analyses the influence of context and process, the relationship between intervention components, and the associated results.

## Supporting information

**S1 Table. Characteristics of the baseline survey sample (2015), stratified by country and network.**
(DOCX)

**S2 Table. Changes in the influencing factors of cross-level clinical coordination (intermediate outcomes) between 2015 and 2017 in the control networks, by country.**
(DOCX)

**S3 Table. Differences in influencing factors of cross-level clinical coordination (intermediate outcomes) in 2015 in the intervention and control networks, by country.**
(DOCX)

**S4 Table. Changes in the experience of cross-level coordination of information and clinical management of care (distal outcomes) between 2015 and 2017 in the control networks, by country.**
(DOCX)

**S5 Table. Differences in experience of cross-level coordination of information and clinical management of care (distal outcomes) in 2015 in the intervention and control networks, by country.**
(DOCX)

**S6 Table. Distribution of influencing factors (intermediate outcomes) of clinical coordination between care levels, intervention and control networks in 2015 and 2017, by country.**
(DOCX)

**S7 Table. Distribution of experience of cross-level coordination of information and clinical management of care (distal outcomes), intervention and control networks in 2015 and 2017, by country.**
(DOCX)

**S8 Table. Perception of coordination between care levels (distal outcomes), in the intervention and control networks in 2015 and 2017, by country.**
(DOCX)

**S9 Table. Summary of positive effects of PAR-based interventions on clinical coordination, by country: Improvement in intermediate and distal outcomes of clinical coordination in IN between 2015 and 2017.**
(DOCX)

## Acknowledgments

The authors are most grateful to the LSC, PP, doctors, managers and other professionals of the networks and research fellows that participated in the study and generously shared their effort, time and opinions, thereby making it possible. We highly appreciate the contributions to this study of the following individuals who, together with the authors of the paper, formed part of the Equity-LA II project led by M.L. Vazquez (mlvazquez@consorci.org): Brazil: Isabella Samico, Cecylia Oliveira, Hylany Almeida, Renata Freitas, Cynthia Resque, Geison Silva, Luciana Dubeux (Instituto de Medicina Integral Prof. Fernando Figueira); Chile: Isabel Guzmán, Patricio Alvarez, Ana-María Oyarce, Andrea Alvarez, Nimsi Pastén, Viviana Rojas, Paola González, Jorge Caro (Universidad de Chile), Isabel Abarca (Servicio de Salud Metropolitano Norte, Santiago de Chile), Maria Eugenia Chadwick (Dirección de Salud Municipio de Recoleta, Santiago), Patricia Espejo (Centro de Diagnóstico y Tratamiento, Hospital San José, Santiago), Mauricio Araya (Dirección de Salud Municipio de Independencia, Santiago), Wilma Hidalgo y Sergio Rojas (Dirección de Salud Municipio de Huechuraba, Santiago); Colombia: Virginia Garcia, Angela-María Pinzón, Heisel-Gloria León, Andrés Gallego, Carol Cardoso, Laura Bejarano, Josefina Chávez, Silvia Ballesteros, Leonardo Gómez, Cesar Santamaría, Carmen Villamizar (Universidad del Rosario); Cristian Cortés, Carolina Larrañaga, Edgar Sanabria (Subred Integrada de Servicios de salud Sur Occidente- Bogotá); Mexico: Angélica-Ivonne Cisneros, Edit Rodríguez, Damián-Eduardo Pérez, Vianey González, Néstor-Iván Cabrera, Daniel Córdoba (Universidad Veracruzana), Uruguay: Sebastián Gadea, Camila Estiben, Luciana Piccardo (Universidad de la República), Graciela García Dirección Región Oeste de ASSE, Cecilia Acosta, Subdirección Región Oeste Litoral de ASSE, María-Noel Ballarini

(Subdirección Región Oeste Centro de ASSE), Verónica Reyes (Dirección Red de Atención Primaria Dpto. de Soriano), Viviana Corona (Dirección Red de Atención Primaria Dpto. de Río Negro), Juan-Pablo Apollonia (Dirección Hospital de Young), Gladys-Yanet Sandes (Dirección Red de Atención Primaria Dpto. de Colonia). Our sincere thanks are also given to Soledad Barria for her reflections throughout the project. We thank Kate Bartlett for her help with the English version of the firstly submitted and revised version of the manuscript; Maria Rubio, Ana Fernández and Carol Cardozo for the external review of the statistical analysis.

## Author Contributions

**Conceptualization:** María-Luisa Vázquez, Ingrid Vargas.

**Formal analysis:** Andrea Miranda-Mendizabal.

**Funding acquisition:** María-Luisa Vázquez, Pamela Eguiguren, Amparo-Susana Mogollón-Pérez, Fernando Bertolotto, Ingrid Vargas.

**Investigation:** María-Luisa Vázquez, Andrea Miranda-Mendizabal, Pamela Eguiguren, Amparo-Susana Mogollón-Pérez, Marina Ferreira-de-Medeiros-Mendes, Julieta López-Vázquez, Fernando Bertolotto, Ingrid Vargas.

**Methodology:** María-Luisa Vázquez, Andrea Miranda-Mendizabal, Ingrid Vargas.

**Software:** Andrea Miranda-Mendizabal.

**Supervision:** María-Luisa Vázquez, Pamela Eguiguren, Amparo-Susana Mogollón-Pérez, Marina Ferreira-de-Medeiros-Mendes, Julieta López-Vázquez, Fernando Bertolotto, Ingrid Vargas.

**Validation:** María-Luisa Vázquez, Pamela Eguiguren, Amparo-Susana Mogollón-Pérez, Ingrid Vargas.

**Visualization:** María-Luisa Vázquez, Ingrid Vargas.

**Writing – original draft:** María-Luisa Vázquez, Andrea Miranda-Mendizabal, Ingrid Vargas.

**Writing – review & editing:** María-Luisa Vázquez, Pamela Eguiguren, Amparo-Susana Mogollón-Pérez, Marina Ferreira-de-Medeiros-Mendes, Julieta López-Vázquez, Fernando Bertolotto, Ingrid Vargas.

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
