## [Decision Letter · Decision Letter 0]

19 Feb 2021

PONE-D-20-36859

Evaluating the effectiveness of participatory action research interventions to improve clinical coordination between care levels: lessons learned from a quasi-experimental study in public healthcare networks in Latin America

PLOS ONE

Dear Dr. Vargas,

Thank you for submitting your manuscript to PLOS ONE. After careful consideration, we feel that it has merit but does not fully meet PLOS ONE’s publication criteria as it currently stands. Therefore, we invite you to submit a revised version of the manuscript that addresses the points raised during the review process.

The manuscript has been evaluated by two reviewers, and their comments are available below. The reviewers have raised a number of concerns that need attention. They request additional information on methodological aspects of the study as well as on the discussion and presentation of the results. Could you please revise the manuscript to carefully address the concerns raised?

We look forward to receiving your revised manuscript.

Kind regards,

Dario Ummarino, Ph.D.

Academic Editor

PLOS ONE

Journal Requirements:

3. One of the noted authors is a group; Equity LA II.

In addition to naming the author group, please list the individual authors and affiliations within this group in the acknowledgments section of your manuscript.

Please also indicate clearly a lead author for this group along with a contact email address.

Reviewers' comments:

Reviewer's Responses to Questions

**Comments to the Author**

1. Is the manuscript technically sound, and do the data support the conclusions?

Reviewer #1: Yes

Reviewer #2: Partly

2. Has the statistical analysis been performed appropriately and rigorously? 

Reviewer #1: Yes

Reviewer #2: Yes

3. Have the authors made all data underlying the findings in their manuscript fully available?

Reviewer #1: Yes

Reviewer #2: No

4. Is the manuscript presented in an intelligible fashion and written in standard English?

Reviewer #1: Yes

Reviewer #2: Yes

5. Review Comments to the Author

Reviewer #1: This paper aims to examine the effects of Participatory Action Research (PAR) Interventions designed to improve coordination of care among primary and specialist clinicians in healthcare networks in 5 Latin American countries. The authors present results from a quasi-experimental evaluation of intermediate interactional and organizational outcomes, as well as physician-reported care coordination outcomes. There are several compelling aspects of this paper, including: the focus on PAR interventions—which are increasingly common in both healthcare and community public health, the multi-country and health system sample, and pairing of established survey outcome measures with qualitative methods. However, several aspects of the paper could be clarified, differences in the health system structure and payment systems among the countries/networks studied could be more clearly described, and preponderance of null results more clearly emphasized.

Please provide a definition or example of “levels of care” early in the paper. This would be helpful to the many PLOS One readers with limited healthcare experience. The terms “primary” and “secondary” care are not mentioned until page 6.

Please also consider adding a very brief summary of the health systems/healthcare payment systems in the participating countries. Alternatively, this information could be added to Supplementary Table 1. Payment policy differences significantly influence coordination methods and time invested in coordination activities. These issues are mentioned briefly in the discussion section but should be addressed earlier.

Please clarify how networks were identified as either an intervention or control network. I understand this was not random assignment, but did network leaders choose if they wanted to be in the intervention group and others did not? Was there some other way the “intervention” networks were selected or identified? Also, the networks differed in the design approaches used, implementation timelines, and types of interventions selected. It would seem that some subgroup analyses of the 3 networks with more similar design and implementation approaches could be useful (i.e., Columbia, Brazil, and Mexico).

I commend the authors for the description of their conceptual model, providing definitions of key study terminology, and using an established survey instrument with prior psychometric evaluation data.

Please clarify the sampling method (i.e., how were participating physicians identified and contacted to participate?) and provide response rates for the baseline and follow-up surveys. Also please clarify the proportion of respondents who completed both baseline and follow-up surveys. It should be clear if this is multiple time-point cross-sectional survey or a true pre-post survey of the same respondents.

Variables section, pg. 9: Perhaps consider using the terminology “intermediate outcomes” and “distal outcomes” rather than “intermediate results” and “final results” to describe these variables.

Clarify why prevalence ratios were used. This suggests that survey item scores were dichotomized.

Sections of Supplementary Table 1 appear to be missing. If specific sections of the table are not applicable to particular countries it would be useful to note “not applicable” in the appropriate cells.

Supplemental table 1 and Table note for Table 1 in the paper: Please clarify intervention 2 in Chile. It is currently described as “cross-level bidirectional visits,” but that description is not clear. Were these in-person multi-disciplinary visits where a patient would see both primary and secondary care providers in the same visit or on the same day? Or something else?

The abstract and discussion section both emphasize the few statistically significant results from the large number of analyses conducted. Conclusions presented in the abstract in particular should be tempered with the limitations of the study and focus on the need for more robust evaluation approaches that are described in the discussion section.

Overall, sentence structure (particularly in the discussion/conclusions section) could be simplified and streamlined. Key points are hidden and hard to follow given the use of very long, complex sentences. This problem is easily correctable, but important if readers are to grasp key conclusions.

E.g., pg. 26, lines 502 to 508: The example below is a single sentence…though it reads like a paragraph…. “However, detecting changes at network/organization level poses challenges that have been extensively highlighted in fields such as community-based evaluation [51,52], such as low statistical power to detect differences because of the limited number of intervention and control organizations/networks that make up the sample, this being due to the financial and logistical difficulties/costs involved in increasing their number; or the need for a longer time frame for implementation, and thus more resources, to achieve greater penetration of the intervention and make an impact at organization/area level.”

Reviewer #2: Thank you for the opportunity to review this manuscript. The authors report the findings of a large, multinational study aimed at evaluating the effectiveness of Participatory Action Research (PAR) at improving coordination between primary and specialty care services at Healthcare Networks in five countries in Latin America (Brazil, Chile, Colombia, Mexico, and Uruguay). Although the work reported by the authors is clearly a Herculean effort, the manuscript as currently written lacks focus and makes it difficult for the reader to understand aspects of the larger study that are being reported on here, and the ensuing conclusions that can be drawn. I focus my comments on three areas of concern:

1. Based on my reading of the manuscript, the authors appear to be evaluating the effectiveness of the PAR approach as an implementation intervention to improve coordination between primary and specialty care. However, the specific components of PAR are that were used in each country were not clearly reported, nor were the resulting changes taken by each country as a result of the PAR approach. In other words, PAR itself does not directly change coordination, it opens the door for the networks to identify the interventions (and implementation strategy) that they will need to change coordination. Those intermediate outcomes were not reported in the paper.

2. In the discussion, the authors present an expansive section on the methodological challenges to participatory action research -- was this intended to serve as the limitations section of their study? If so, the section does not achieve its goal, as it does not report on the specific challenges of the study and how they were mitigated, but rather talks about general methodological challenges. If the intent of that section was not to serve as the limitations section, then the limitations section of the study is missing. Either way, specifics about the limitations of the study and attempt to mitigate these limitations is missing from the manuscript.

3. There is a lot of confusion in the methods section in reporting of what was actually done. It took me to the end of the methods section to realize and understand that interactional factors, organizational factors, and the various clinical coordination factors were actually subscales in the COORDENA questionnaire. There's also talk throughout the paper about a qualitative study that is not reported in the manuscript. Either delete the language about the qualitative study, or re-ported in greater detail here.

4. This is more of a presentation issue: the authors provide in luxury of detailed tables reporting the various comparisons, both in the main text at a supplementary material. However, is the number of comparisons is so great that for the reader it becomes very cumbersome and unwieldy to be able to discern what was actually found and what significant improvements are worthy of focus. A summary table might be a good way to make it easier for the reader to understand what was most relevant and most important.

Other, smaller matters:

1. Lines 178-194 belong better in the introduction

2. Line 176: "This study is oriented by a comprehensive conceptual framework and care coordination of cross care levels..." To what framework are you referring? Please specify.

6. PLOS authors have the option to publish the peer review history of their article (what does this mean?). If published, this will include your full peer review and any attached files.

Reviewer #1: No

Reviewer #2: No

---

## [Author Response · Author response to Decision Letter 0]

17 Mar 2021

Editor’s comments

We have revised the manuscript to ensure that it meets PLOS ONE’s style requirements.

2. We note that you have indicated that data from this study are available upon request. PLOS only allows data to be available upon request if there are legal or ethical restrictions on sharing data publicly. In your revised cover letter, please address the following prompts:

b) If there are no restrictions, please upload the minimal anonymized data set necessary to replicate your study findings as either Supporting Information files or to a stable, public repository and provide us with the relevant URLs, DOIs, or accession numbers. Please see http://www.bmj.com/content/340/bmj.c181.long for guidelines on how to de-identify and prepare clinical data for publication. For a list of acceptable repositories, please see http://journals.plos.org/plosone/s/data-availability#loc-recommended-repositories. We will update your Data Availability statement on your behalf to reflect the information you provide.

We have held an in-depth discussion on the possibility of publishing the data on which this study is based with the Data Protection Officer of the project, and have concluded that there are no ethical or legal restrictions on sharing the data set publicly, since it is fully anonymous and presents minimal risk to the confidentiality of study participants.

We have updated our Data Availability statement to reflect this information and provided the URL of the public repository from which the data set can be downloaded.

3. One of the noted authors is a group; Equity LA II. In addition to naming the author group, please list the individual authors and affiliations within this group in the acknowledgments section of your manuscript. Please also indicate clearly a lead author for this group along with a contact email address.

We have listed the individual authors and affiliations within Equity-LA II, which contributed to the development of this study, as well as the lead author (María-Luisa Vázquez), in the acknowledgments section of the manuscript. 

Reviewer's Responses to Questions

1. Is the manuscript technically sound, and do the data support the conclusions?

Reviewer #1: Yes

Reviewer #2: Partly

We have rewritten the conclusions in the abstract and discussion sections of the manuscript to emphasize the results more clearly, as indicated below. 

2. Have the authors made all data underlying the findings in their manuscript fully available?The PLOS Data policy requires authors to make all data underlying the findings described in their manuscript fully available without restriction, with rare exception (please refer to the Data Availability Statement in the manuscript PDF file). The data should be provided as part of the manuscript or its supporting information, or deposited to a public repository. 

Reviewer #1: Yes

Reviewer #2: No

As stated under Editors’ comments 2, the data are now fully available from a public repository.

Reviewer 1

Firstly, we would like to thank the reviewer for their comments and the suggestions received, which we have incorporated into the revised manuscript.

1. Please provide a definition or example of “levels of care” early in the paper. This would be helpful to the many PLOS One readers with limited healthcare experience. The terms “primary” and “secondary” care are not mentioned until page 6.

“Levels of care” refers to primary care and secondary care, distinguished according to the level of complexity. We have now clarified the term “levels of care” earlier in the paper (line 91, page 4)

2. Please also consider adding a very brief summary of the health systems/healthcare payment systems in the participating countries. Alternatively, this information could be added to Supplementary Table 1. Payment policy differences significantly influence coordination methods and time invested in coordination activities. These issues are mentioned briefly in the discussion section but should be addressed earlier.

Following the reviewer’s recommendation, we have added a brief summary of the characteristics of the public healthcare sub-systems in the participating countries, including the health system funding and payment mechanisms). This will contribute to a better understanding of the context and the economic incentives to take part in the participatory process and the adoption of interventions (line 216, page 8).

3. Please clarify how networks were identified as either an intervention or control network. I understand this was not random assignment, but did network leaders choose if they wanted to be in the intervention group and others did not? Was there some other way the “intervention” networks were selected or identified?

The selection of networks in each country was not random, but was done by applying the following selection criteria: a) provision of a continuum of services to a defined population including at least primary and secondary care; b) mainly in urban areas of low or medium-low socioeconomic status; and c) willingness to participate and implement the designed interventions. Therefore, the selected intervention network had to be willing to implement the intervention, as stated in the last criterion. The participation of the networks was voluntary and there were no refusals to participate. The final decision was made by the researchers together with the managers of the networks.

We have added this information in the section on study design and study areas (line 248, page 9).

4. Also, the networks differed in the design approaches used, implementation timelines, and types of interventions selected. It would seem that some subgroup analyses of the 3 networks with more similar design and implementation approaches could be useful (i.e.,Columbia, Brazil, and Mexico).

As the reviewer points out, the participatory process, the type and characteristics of the interventions implemented, and the contextual factors that influenced the process differed across the five experiences. These aspects have been analyzed comparatively in two previous publications(1,2) and we are currently preparing a third article that focuses on a comparative analysis of the PAR processes for the design of the interventions in the study countries. Furthermore, several papers are underway that analyze in greater depth and detail the design and implementation process carried out in each country on an individual basis. 

With regard to potential subgroups of analysis, although we acknowledge that there are some similarities, these are not enough to clearly establish groups of countries for comparative analysis: firstly, the experiences may have certain elements in common in terms of the participatory process, but they differ in several others, particularly in the type of interventions that were implemented and in the contextual factors that influenced them; and secondly, the number of study countries (five) is too small to be able to carry out a subgroup analysis.

5. I commend the authors for the description of their conceptual model, providing definitions of key study terminology, and using an established survey instrument with prior psychometric evaluation data.

We would like to thank the reviewer for the comments on the description of the conceptual model and survey instrument.

6. Please clarify the sampling method (i.e., how were participating physicians identified and contacted to participate?) and provide response rates for the baseline and follow-up surveys. Also please clarify the proportion of respondents who completed both baseline and follow-up surveys. It should be clear if this is multiple time-point cross-sectional survey or a true pre-post survey of the same respondents. 

Following the reviewer's recommendation, we have added information about how the participants were contacted, and the response rate for both the baseline and the evaluation surveys in the data collection section. The selection of the participants for each survey was carried out based on the list of physicians working at the center at the time of survey who met the selection criteria. Therefore the participating physicians were not necessarily the same for both surveys (line 262, page 9)

A question was included in the evaluation survey (in some countries) asking about participation in the first survey (self-reported), since the survey was anonymous. We didn’t consider it appropriate to include this variable in the analysis, nor to report it in the article, because a recall bias was observed. 

7. Variables section, pg. 9: Perhaps consider using the terminology “intermediate outcomes” and “distal outcomes” rather than “intermediate results” and “final results” to describe these variables. 

Following the reviewer’s recommendation, we have replaced the term intermediate/final results with intermediate/distal outcomes consistently throughout the manuscript. 

8. Clarify why prevalence ratios were used. This suggests that survey item scores were dichotomized. 

The outcome variables were dichotomized by merging original answer options into: high (always/often=1) or low (sometimes/rarely/never=0). We have included this information in the methods section (line 331, page 10).

Thanks to the reviewer's comment, we realized that in the supplementary tables we did not include the corresponding table footer to indicate the grouping of the responses shown in the table. We have now added it.

9. Sections of Supplementary Table 1 appear to be missing. If specific sections of the table are not applicable to particular countries it would be useful to note “not applicable” in the appropriate cells.

Indeed, specific sections of the table are not applicable to particular countries. We have now added “NA = not applicable” in the corresponding cells (Table S1).

10. Supplemental table 1 and Table note for Table 1 in the paper: Please clarify intervention 2 in Chile. It is currently described as “cross-level bidirectional visits,” but that description is not clear. Were these in-person multi-disciplinary visits where a patient would see both primary and secondary care providers in the same visit or on the same day? Or something else?

The cross-level bidirectional visits were carried out by PC and SC teams. That is to say, both teams would visit the centres where the team of the other level worked. The objective was to promote “mutual knowledge” between professionals across care levels (i.e. knowledge of their work environment, getting to know each other in person, and establishing ties). Patients were not involved. The visits formed part of the network induction program for newly appointed personnel, alongside graphic and audiovisual material providing information on the network's characteristics (operational units, equipment, workers, population served, etc.) and activities. The goal of the induction program was to promote a common identity and shared vision in the healthcare network. The cross-level visits lasted 4 hours (half a work day), and a total of 4 visits were carried out during the project. 

The details of the interventions are provided and analyzed in the article on the design of the interventions that we are currently preparing. However, we have added some more information in the Tables 1 and S1, and text (line 396, page 12) to clarify the contents of this intervention.

11. The abstract and discussion section both emphasize the few statistically significant results from the large number of analyses conducted. Conclusions presented in the abstract in particular should be tempered with the limitations of the study and focus on the need for more robust evaluation approaches that are described in the discussion section.

We have revised the conclusions in the abstract and the introduction to the discussion section to explain that: 1) the observed improvements refer to some items of clinical coordination; 2) there are differences between countries in the outcomes obtained; 3) a longer evaluation time frame is required to detect more quantifiable changes (line 65, page 3 and line 501-9, page 23). However, we would like to highlight, as indicated in the discussion, the importance of the achievements considering the limited time frame of this intervention evaluation and their consistency with the expected results of these interventions, which were selected by the health professionals themselves to improve their daily practice. 

12. Overall, sentence structure (particularly in the discussion/conclusions section) could be simplified and streamlined. Key points are hidden and hard to follow given the use of very long, complex sentences. This problem is easily correctable, but important if readers are to grasp key conclusions.

E.g., pg. 26, lines 502 to 508: The example below is a single sentence…though it reads like a paragraph…. “However, detecting changes at network/organization level poses challenges that have been extensively highlighted in fields such as community-based evaluation [51,52], such as low statistical power to detect differences because of the limited number of intervention and control organizations/networks that make up the sample, this being due to the financial and logistical difficulties/costs involved in increasing their number; or the need for a longer time frame for implementation, and thus more resources, to achieve greater penetration of the intervention and make an impact at organization/area level.”

Following the reviewer's recommendations, we have revised the discussion and conclusions to simplify the sentence structures and clarify the main messages. 

Reviewer 2

First of all, we would like to thank the reviewer for their comments and the suggestions received, which we have incorporated into the revised manuscript. 

1. Based on my reading of the manuscript, the authors appear to be evaluating the effectiveness of the PAR approach as an implementation intervention to improve coordination between primary and specialty care. However, the specific components of PAR are that were used in each country were not clearly reported, nor were the resulting changes taken by each country as a result of the PAR approach. In other words, PAR itself does not directly change coordination, it opens the door for the networks to identify the interventions (and implementation strategy) that they will need to change coordination. Those intermediate outcomes were not reported in the paper.

Indeed, as stated by the reviewer, “the PAR itself does not directly change coordination, but it opened the door for the networks to identify the interventions” needed, and “the intermediate results are not reported here”.

In this regard, we would like to highlight that this study is only one part of a wider implementation research project (Equity-LA II) (3), which adopts a PAR approach at different levels and, as the reviewer highlights, particularly for the selection, design and implementation of specific interventions to improve cross-level clinical coordination. To evaluate the implemented interventions, this research combines qualitative data collection methods focusing on the design and implementation process of the interventions (5) with quantitative methods to analyze the effectiveness of the implemented intervention in terms of improvements in doctors’ care coordination experience at network level. The latter is the focus of this paper. 

As the results of these exhaustive evaluations are so extensive, both for each country case and across countries, their analysis has been broken down into several different publications. The comparative analysis of the qualitative evaluation results focused on: 1) analysis of contextual factors and the PAR process that influenced the implementation of the interventions(1); 2) the results of the interventions in improving coordination between levels of care as perceived by the directly involved participants (2); and 3) analysis of the participatory process for the design of the interventions (in preparation). In other words, the intermediate outcomes of the PAR process indicated by the reviewer were evaluated with the participants of the interventions using qualitative research methods. The results have been published in two previous articles (2,5), and the third one is underway. These findings have been taken into account for the interpretation of the results concerning the effect of the intervention on cross-level care coordination in the discussion of the submitted manuscript. 

With regard to the effect of the PAR process on care coordination, as mentioned above, the participatory process was aimed at designing and implementing interventions to improve coordination in the networks. In this process, the professionals who participated selected significant coordination problems in their daily practice, and subsequently selected and designed interventions to address these problems (6). Therefore, the objectives of the interventions were to improve care coordination between care levels. One of the results of the implementation process evaluation is that the participatory process, in some countries, not only contributed to a higher uptake of the interventions (by fostering professionals’ interest and motivation to participate), but also improved the interactional factors that influence coordination between levels of care (personal relationships, mutual trust and training of professionals) (2,5). In short, with differences between countries, the PAR process contributed directly to improving coordination between the participants of the interventions, as discussed in lines 570-7, page 25.

In order to clarify these points, we have provided further information about the qualitative evaluation and the references to related publications in the introduction of the article (lines 169- 182, page 6-7). We have also revised the discussion to better explain the contribution of the PAR process to improving coordination between levels (lines 570-7, page 25). 

2. In the discussion, the authors present an expansive section on the methodological challenges to participatory action research -- was this intended to serve as the limitations section of their study? If so, the section does not achieve its goal, as it does not report on the specific challenges of the study and how they were mitigated, but rather talks about general methodological challenges. If the intent of that section was not to serve as the limitations section, then the limitations section of the study is missing. Either way, specifics about the limitations of the study and attempt to mitigate these limitations is missing from the manuscript.

This section details the methodological difficulties encountered over the course of this study and the strategies used to deal with them. Most of the difficulties are inherent to the nature of the PAR method, and have been reported by other community-based evaluations, but not in the field of health services evaluation. For this reason, it seemed important to us to include more generalized reflections and make recommendations for future evaluations of the effectiveness of PAR interventions in health services. We agree with the reviewer that some of these reflections are too general, and as a result, we have change the title of the section and re-rewritten the section to clarify the methodological difficulties found in this study, the strategies used to tackle them, and the recommendations for future evaluations (lines 639-826, page 27-28).

3. There is a lot of confusion in the methods section in reporting of what was actually done. It took me to the end of the methods section to realize and understand that interactional factors, organizational factors, and the various clinical coordination factors were actually subscales in the COORDENA questionnaire. There's also talk throughout the paper about a qualitative study that is not reported in the manuscript. Either delete the language about the qualitative study, or re-ported in greater detail here.

We have revised the methods section to make it clearer that the variables related to the final and intermediate outcomes of the interventions were collected using the COORDENA questionnaire. However, these are not subscales of the questionnaire, but items that are treated individually. Although detailed information on the design, structure, and validation of the questionnaire has already been published (4,7), we have also added some more information regarding the structure of the questionnaire (line 307, page 10). 

With regard to the last point, as mentioned above, this study is part of a wider implementation research project (3) which combines qualitative research methods to evaluate the design and implementation process of the interventions (5) with quantitative methods to analyze their effectiveness in terms of care coordination. This comprehensive hybrid approach is increasingly adopted to simultaneously evaluate the clinical effectiveness of an intervention and the implementation process (8,9), to provide information on contextual effectiveness. In this paper, the qualitative evaluation of the implementation process, the factors that influenced it and the results perceived by participants, complements and facilitates interpretation of the findings of the effectiveness analysis. It helps to identify and understand the factors of success or failure for the intervention related to the context and the process, the mechanisms by which the intervention has contributed to the associated results, and other outcomes identified by the participants of the interventions but not detected in the evaluation in the whole network of health services. For this reason, they have been taken into account in the interpretation of the results in the discussion of the article.

In the introduction to the article we have now provided more information about the qualitative evaluation, references to previous publications, and clarified the results presented in this article (line 168, page 6). 

4. This is more of a presentation issue: the authors provide in luxury of detailed tables reporting the various comparisons, both in the main text at a supplementary material. However, is the number of comparisons is so great that for the reader it becomes very cumbersome and unwieldy to be able to discern what was actually found and what significant improvements are worthy of focus. A summary table might be a good way to make it easier for the reader to understand what was most relevant and most important.

Following the reviewer's recommendations, we have created a supplementary table to summarize only the statistically significant changes found in the intermediate and final outcomes of the interventions in each country (Table S6). 

5. Lines 178-194 belong better in the introduction

We have moved the description of the conceptual framework to the introduction section to make it more visible (line 138, page 5).

6. Line 176: "This study is oriented by a comprehensive conceptual framework and care coordination of cross care levels..." To what framework are you referring? Please specify.

Our study is based on a previously developed conceptual framework for the analysis of care coordination across care levels, which comprehensively addresses the different types and dimensions of clinical coordination, the care coordination mechanisms to improve them, and the factors that influence care coordination (best described in Vázquez et al. (2015)(10)). This theoretical framework was initially developed based on an extensive literature review (11–13) and has been improved over the years through its application in a number of quantitative and qualitative studies in Latin America and Catalonia (Spain)(4,14–17). 

Some of the main elements of this framework are summarized in the introduction section of this article (definitions and types of care coordination, types of care coordination mechanisms and factors). In order to clarify this further, a subtitle has been added within this section, as well as a sentence explaining the conceptual framework and the corresponding references (line 138, page 5).

1. Vargas I, Vázquez ML. Understanding the factors influencing the implementation of participatory interventions to improve care coordination. An analytical framework based on an evaluation in Latin America. Heal Policy Plan Approv. 2020; 

2. Vargas I, Eguiguren P, Mogollón-Pérez A-S, Samico I, Bertolotto F, López-Vázquez J, et al. Can care coordination across levels be improved through the implementation of participatory action research interventions? Outcomes and conditions for sustaining changes in five Latin American countries. BMC Health Serv Res. 2020;20(1):941. 

3. Vazquez ML, Vargas I, Unger JP, De Paepe P, Mogollon-Perez AS, Samico I, et al. Evaluating the effectiveness of care integration strategies in different healthcare systems in Latin America: the EQUITY-LA II quasi-experimental study protocol. BMJ Open [Internet]. 2015;5(7):e007037. Available from: http://www.ncbi.nlm.nih.gov/pubmed/26231753

4. Vazquez ML, Vargas I, Garcia-Subirats I, Unger J-PP, de Paepe P, Mogollón-Pérez AS, et al. Doctors’ experience of coordination across care levels and associated factors. A cross-sectional study in public healthcare networks of six Latin American countries. Soc Sci Med. 2017;182:10–9. 

5. Vargas I, Vázquez M-LL, Eguiguren P, Mogollón-Pérez A-S, Bertolotto F, Samico I, et al. Understanding the factors influencing the implementation of participatory interventions to improve care coordination. An analytical framework based on an evaluation in Latin America. Health Policy Plan. 2020;35(8):962–72. 

6. Vargas I, Eguiguren P, Mogollón A, López J, Samico I, Bertolotto F, et al. [Participatory design of interventions to improve clinical coordination in Latin America]. Gac Sanit. 2019;33(SC). 

7. Vargas I, Garcia-Subirats I, Mogollón-Pérez A-S, Ferreira-de-Medeiros-Mendes M, Eguiguren P, Cisneros A-I, et al. Understanding communication breakdown in the outpatient referral process in Latin America: a cross-sectional study on the use of clinical correspondence in public healthcare networks of six countries. Health Policy Plan [Internet]. 2018;czy016–czy016. Available from: http://dx.doi.org/10.1093/heapol/czy016

8. Curran GM, Bauer M, Mittman B, Pyne JM, Stetler C. Effectiveness-implementation hybrid designs: combining elements of clinical effectiveness and implementation research to enhance public health impact. Med Care [Internet]. 2012;50(3):217–26. Available from: http://www.ncbi.nlm.nih.gov/pubmed/22310560

9. Damschroder LJ, Aron DC, Keith RE, Kirsh SR, Alexander JA, Lowery JC. Fostering implementation of health services research findings into practice: A consolidated framework for advancing implementation science. Implement Sci. 2009;4(1). 

10. Vázquez ML, Vargas I, Unger JP, De Paepe P, Mogollón-Pérez AS, Samico I, et al. Evaluating the effectiveness of care integration strategies in different healthcare systems in Latin America: The EQUITY-LA II quasi-experimental study protocol. BMJ Open. 2015;5(7):1–10. 

11. Vázquez ML, Vargas I, Unger JP, Mogollón AS, Ferreira da Silva M, De Paepe P. Integrated healthcare network in Latin America: towards a conceptual framework for analysis. Pan Am J Public Heal. 2009;26(4):360–7. 

12. Terraza-Núñez R, Vargas I, Vázquez ML. [Coordination among healthcare levels: systematization of tools and measures]. Gac Sanit [Internet]. 2006;20(6):485–95. Available from: http://www.ncbi.nlm.nih.gov/pubmed/17198628

13. Vázquez Navarrete ML, Vargas Lorenzo I, Mogollón-Pérez AS, Ferreira da Silva MR, Unger J-P, de Paepe P. Redes integradas de servicios de salud en Colombia y Brasil: Estudio de casos. Editorial Universidad del Rosario; 2018. 

14. Vargas I, Mogollón-Pérez AS, De Paepe P, Ferreira Da Silva MR, Unger JP, Vázquez ML. Barriers to healthcare coordination in market-based and decentralized public health systems: A qualitative study in healthcare networks of Colombia and Brazil. Health Policy Plan. 2016; 

15. Esteve-Matalí L, Vargas I, Sánchez E, Ramon I, Plaja P, Vázquez M. Do primary and secondary care doctors have a different experience and perception of cross-level clinical coordination? Results of a cross-sectional study in the Catalan National Health System (Spain). BMC Fam Pract. 2020;21(1). 

16. Aller MB, Vargas I, Coderch J, Calero S, Cots F, Abizanda M, et al. Doctors’ opinions on clinical coordination between primary and secondary care in the Catalan healthcare system. Gac Sanit. 2019;33(1):66–73. 

17. Aller MB, Vargas I, Coderch J, Calero S, Cots F, Abizanda M, et al. Development and testing of indicators to measure coordination of clinical information and management across levels of care. BMC Health Serv Res. 2015;

---

## [Decision Letter · Decision Letter 1]

23 Jun 2021

PONE-D-20-36859R1

Evaluating the effectiveness of participatory action research interventions to improve clinical coordination between care levels: lessons learned from a quasi-experimental study in public healthcare networks in Latin America

PLOS ONE

Dear Dr. Vargas,

Thank you for submitting your manuscript to PLOS ONE. After careful consideration, we feel that it has merit but does not fully meet PLOS ONE’s publication criteria as it currently stands. Therefore, we invite you to submit a revised version of the manuscript that addresses the points raised during the review process.

The manuscript was independently reviewed by two reviewers, one of them familiar with the original submission and one new to the manuscript, in addition to my own reading of the piece. Both reviewers agree that the manuscript is highly responsive to the previous reviewers' concerns, and that much can be learned from the execution of this study.  However, both reviewers' (particularly Reviewer 3) comments focus on the clarity of the presentation of the work reported. I agree with their assessment -- the manuscript needs to be revised in order to help the reader clearly follow the main message from the study findings, and to clearly understand how this set of findings fits in the context of the larger study. Reviewer 3 in particular provides detailed and thoughtful suggestions on this matter so that the paper is easier to follow while still providing sufficient detail to attempt replication.  Although it is not required that each individual suggestion be implemented, I strongly suggest you consider carefully the underlying problems the reviewer's suggestions are trying to address, and address each underlying problem as a whole.  

Please also pay specific attention to Reviewer 3's concern regarding regarding multiple statistical comparisons. Given the considerable number of comparisons presented, Reviewer 3's request for statistical correction (e.g., Bonferroni, Tukey, Scheffe) is warranted.  

I realize the requested revisions represent a significant investment of time.  If you choose not to revise, we wish you well in finding another venue and hope that you will consider us again in the future. If you do choose to revise, we look forward to your new draft.

We look forward to receiving your revised manuscript.

Kind regards,

Sylvia J Hysong

Academic Editor

PLOS ONE

Reviewers' comments:

Reviewer's Responses to Questions

**Comments to the Author**

1. If the authors have adequately addressed your comments raised in a previous round of review and you feel that this manuscript is now acceptable for publication, you may indicate that here to bypass the “Comments to the Author” section, enter your conflict of interest statement in the “Confidential to Editor” section, and submit your "Accept" recommendation.

Reviewer #1: (No Response)

Reviewer #3: (No Response)

2. Is the manuscript technically sound, and do the data support the conclusions?

Reviewer #1: Yes

Reviewer #3: Partly

3. Has the statistical analysis been performed appropriately and rigorously? 

Reviewer #1: Yes

Reviewer #3: No

4. Have the authors made all data underlying the findings in their manuscript fully available?

Reviewer #1: Yes

Reviewer #3: Yes

5. Is the manuscript presented in an intelligible fashion and written in standard English?

Reviewer #1: Yes

Reviewer #3: Yes

6. Review Comments to the Author

Reviewer #1: I thank the authors for their thoughtful consideration of my prior comments, many of my prior concerns were addressed in the revised manuscript. My only remining concern is the substantial discussion of implementation factors in the discussion section that came from previously reported qualitative data. I fully understand that these previously published qualitative data help provide context and insight regarding the quantitative results reported in this paper. However, parts of the current discussion section (e.g., pg 25, line 460 through pg 27, line522) appears to largely re-state findings previously described in Vargas et al., (2020, BMC HSR). Rather than restate these qualitative findings verbatim, I encourage the authors to condense this section and point readers to the prior paper.

Reviewer #3: I'm forced to cut-and-paste comments here but it will be much easier to follow the attached formatted file.

Thank you for the opportunity to review this manuscript that describes one piece of a very ambitious project. The aim of this study was to evaluate the effectiveness of PAR interventions in improving clinical coordination between care levels.

This paper has already been through one round of reviews and I am joining as a reviewer in the second round. Unfortunately, I must highlight issues that really need to be addressed, that were not highlighted in the first round.

First, I want to congratulate the whole team for this large project that spanned five (actually, six) countries using a challenging, complex intervention (PAR) with complex methods. There is a LOT to cover in this paper and the authors have clearly improved the paper from the first round. However, key issues must be addressed. These suggestions will help make this paper much clearer and should help strengthen the framing of results by being more transparent about methods and rationale. One note: it was difficult for me to track responses to prior reviews because line and page numbers did not match the content described in each response.

MAJOR REVISIONS

1. This is an easy fix but a major problem: the authors use the term “intervention” throughout. At times “intervention” seems to refer to the PAR approach used and other times, it seems to refer to the “solution” chosen by each country. If my understanding is accurate, I suggest this terminology:

a. Identified gap/problem: this is the “problematized” issue identified by baseline data(?) [line 165] within each IN/country

b. Chosen solution: this is the “selected intervention(s)” [line 165] within the INs

c. PAR Approach: three general approaches were used by the five countries [lines 166-170] – but each country followed very different PAR approaches that don’t seem to align with the 3 types (ref Table S1).

d. Note, throughout, five countries are referenced but in Line 185, six countries are referenced. At Line 170, it states that Argentina was dropped because the “selected intervention” was not implemented. The authors need to provide better justification of dropping this country. Using an “intent to treat” approach, Argentina should be included. However, the authors could frame this analysis as including only “completers” or “active participants.” Either way, the failure of the Argentinian IN to implement their chosen intervention (aka chosen solution) needs to be discussed in limitations. Previously published findings all seem to have included all six countries.

2. In response to the first item from Reviewer #2, the authors present a nice summary of previously published findings that maps out the work already done and how it informs the current manuscript. I suggest reworking the Introduction to better walk readers through what already has been reported and what this manuscript adds. The basic story arc seems to be this: you’ve reported on the PAR approaches, the “problems/gaps” and “solutions” chosen by each IN within each country and now, your aim is “to evaluate the effectiveness of PAR interventions in improving clinical coordination between care levels.”

a. It would be more accurate to state that your aim is “to evaluate the effectiveness of PAR approaches on indicators (or perceptions) of clinical coordination between care levels.” � you are reporting perceptions/self-reported measures via the COORDENA survey instrument, not actual measures of “coordinated care.”

b. Consider moving Table S1 to the main manuscript. By restructuring this table, it can quickly “tell the story” of each country. Here’s a suggested layout:

i. Title � Table 1: Description of Problems, Solutions, and PAR Approaches used by Country

ii. The current Table S1 has this footnote: “More details on the characteristics of each intervention in: http://www.equity-la.eu/en/publicaciones.php?t=PR” but this link goes to presentations without a description of “interventions” (and consider noting that content is in Spanish)

iii. Rows:

1. Problem(s) (e.g., lack of coordination for diabetes patients)

2. Solution(s) (e.g., create and implement guidelines shared by PC and SC providers that focus on essential practices and care pathways)

3. PAR Approach – Joint Meetings:

a. Description (e.g., Discuss mental health care cases face-to-face

b. Who (e.g., PC teams and psychiatrists)

c. Dose (e.g., 11 design(?) sessions with PC and SC providers)

d. Duration (e.g., 7 months)

4. PAR Approach – Virtual Consultations

a. Description (e.g., Asynchronous virtual consultations in Mental Health via email)

b. Who (e.g., PC doctors and psychiatrists)

c. Dose (e.g., 11 consultations)

d. Duration (e.g., 3 months)

5. PAR approach – Other

a. Description

b. Who

c. Dose

d. Duration

iv. Columns: list the five countries

c. The text description of the 3 types of PAR used (in Lines 166-170) is confusing. Please clarify how these three general approaches align with the content of Table S1/new Table 1 outlined above.

3. The structure and content of all of the Tables needs to be reworked.

a. Table 1 lists “demographic” descriptions of networks and providers within each of the five participating countries. This table is very similar to Table S2 and yet the numbers are different. This needs to be resolved.

b. Table 1 is labeled as “Evaluation Survey (2017)” and yet the footnote in Table 1, for “***” and “**” says that these symbols indicate differences at baseline. These are for differences between IN and CN…at baseline? This Table should focus only on baseline differences, not a mix of both time points.

c. Table 1 was suggested by a reviewer but there is no rationale for the choice of data to show in this table – e.g., sex and age of providers does not seem to have any relationship to the aim. I suggest moving this table to the Supplement except for a few of the rows:

i. From Table 1, toward the bottom: I suggest listing “Enough time during consultation…” and “Knowledge of the Intervention” and “use of/participation in the intervention” be moved to a separate Table to present in Results. These lines are inferred in the Discussion when explaining possible reasons for differences and lack of differences in the care coordination indicators.

d. Tables 2 and 3 are very confusing. Methods need to better describe what “PR” is and why that was chosen as your main outcome. AND provide rationale for showing both IN versus CN differences in Table 2 PLUS showing PRs for reversed order in time (2015 vs 2017). I’m seeing 100 paired comparisons across these two tables! With no adjustment for repeated comparisons, statistically. See below for more on Methods.

i. This paper should be reviewed by a statistician and/or better framed to avoid reaching conclusions that are beyond what the data can support

4. The Methods Section needs to be significantly strengthened.

a. Start with the description of your care coordination framework.

i. Readers need more information on the components of care coordination as conceptualized within the COORDENA questionnaire. Readers are not familiar with COORDENA. It seems that CC (care coordination) comprises 1) interactional factors, 2) Organizational Factors; 3) Coordination of information; 4) Consistency of care ; and 5) patient follow-up

1. You conceptualize 1-2 as “intermediate” and 3-5 as “distal.” More accurate labels might be determinants of CC versus perceptions of CC…or other similar labeling

ii. There is an implied causal/influential pathway: “Enough time during consultation…,” “Knowledge of the PAR intervention,” and “Use of/participation in PAR” (listed in Table 1) may influence Interactional and Organizational factors (listed in Tables 2-3) which may lead to (influence) Coordination of Information, Consistency of care, and Patient follow-up (listed in Tables 4-5). Please describe your hypothesized pathway and intent to explore associations along this pathway.

b. Study Design

i. You characterize this study as “quasi-experimental” which not a helpful label. It would be more accurate to describe this as a prospective non-randomized trial. You pre-selected INs and CNs and then tested the effects of PAR on perceptions of CC (care coordination). Each network identified different problems/gaps and solution(s) and each used a different mix of PAR approaches

c. Study Population and Sample

i. Your unit of analysis seems to be “Country (or Network(?))” – an intervention network (IN) and control network (CN), which are each in separate countries. State this.

ii. Within each country/network, you sampled doctors at two levels: 1) primary; and 2) secondary. Define what types of care are included in each of these levels. These providers are clustered by network – this forms the basis for your outcomes.

d. Primary outcome

i. Hundreds of paired comparisons are presented. Any statistician (and I am not a statistician) will warn against multiple comparisons – and any conclusions based on a seeming random assortment of “significance” from among the hundreds.

ii. A single primary analysis is needed. The detailed paired comparisons can be presented (in a more limited and thoughtful way) to help explain (negative?) results of your primary analysis.

iii. Consider a table that shows baseline results for IN and CNs overall, with ranges across the countries.

iv. What is “PR” – how is it computed? This seems to be your primary outcome. However, it is not used appropriately. A single set of ratios – or difference in difference scores? – should be listed for each of the measures

1. Regarding measurement: will the COORDENA measures support “scale-level” averages for each of the 5 dimensions (e.g., organizational factors as a single averaged score, instead of comparing every item (e.g., a single indicator for organizational factors instead of 2 separate items))?

5. When the measures are better defined as suggested above, along with an appropriate analytic plan, write up results that follow that flow: present primary outcomes followed by exploratory analyses to help explain findings.

a. Table S6 is a nice summary of “significant” findings but combining “significance” for IN vs CN comparison with baseline versus follow-up within IN-only, into the same table, is not theoretically justified.

7. PLOS authors have the option to publish the peer review history of their article (what does this mean?). If published, this will include your full peer review and any attached files.

Reviewer #1: No

Reviewer #3: No

---

## [Author Response · Author response to Decision Letter 1]

30 Aug 2021

Editor’s comments

1. The manuscript was independently reviewed by two reviewers, one of them familiar with the original submission and one new to the manuscript, in addition to my own reading of the piece. Both reviewers agree that the manuscript is highly responsive to the previous reviewers' concerns, and that much can be learned from the execution of this study. However, both reviewers' (particularly Reviewer 3) comments focus on the clarity of the presentation of the work reported. I agree with their assessment -- the manuscript needs to be revised in order to help the reader clearly follow the main message from the study findings, and to clearly understand how this set of findings fits in the context of the larger study. Reviewer 3 in particular provides detailed and thoughtful suggestions on this matter so that the paper is easier to follow while still providing sufficient detail to attempt replication. Although it is not required that each individual suggestion be implemented, I strongly suggest you consider carefully the underlying problems the reviewer's suggestions are trying to address, and address each underlying problem as a whole. 

Please also pay specific attention to Reviewer 3's concern regarding regarding multiple statistical comparisons. Given the considerable number of comparisons presented, Reviewer 3's request for statistical correction (e.g., Bonferroni, Tukey, Scheffe) is warranted. 

We would like to thank the Editor and reviewers for their comments, which have allowed us to improve the manuscript. It was a pity that we did not receive these comments at the same time as the previous ones, because we would have gained a better overview of what needed to be modified and/or strengthened in the manuscript. We have answered all comments point by point and made the necessary changes (or given a rationale for not doing so) very carefully and thoroughly, whilst trying to avoid contradictions with the changes suggested in the first round. 

Following the recommendations of the editor and reviewers, we have made an effort to explain better how the study is embedded in the larger study, given more detail on the design and implementation process of the interventions for their replicability in other contexts, and provided more information for readers to understand the rationale of the analysis performed and the study hypothesis behind it. We have also simplified the presentation of the results in the tables, to make them more coherent with the data analysis, which we think will give more clarity to the findings. Finally, with regard to correcting for multiple statistical comparisons, a review by two external statisticians confirmed that this was not an appropriate method for this study, firstly, due to the statistical problems it poses [1] and secondly, because this kind of test is not considered adequate for quasi-experimental (controlled before-and-after) studies like this one, which have pre-planned and justifiable contrasts of hypothesis [2] for analysing the effectiveness of interventions based on their expected outcomes and on theoretical assumptions supported by the theoretical framework [3–5]. Instead, as recommended, the descriptions of the rationale for the analysis performed and of the study hypothesis were improved.

2. We note you state: "All data files are available for download from the repository of the Equity-LA II project (http://www.equity-la.eu/en/publicaciones.php?t=IS)." Before we can proceed, please address the following prompts:

a. Are these third party data (i.e., data not owned or collected by the author(s))?

b. If these are indeed third party data, please explain how others can access these datasets and confirm that others would be able to access these data in the same manner as the authors. Please also confirm that the authors did not have any special access privileges that others would not have.

c. If these are not third party data but there are ethical or legal restrictions on sharing a de-identified data set, please explain them in detail (e.g., data contain potentially identifying or sensitive information) and who has imposed them (e.g., a governmental body, an ethics committee, etc.). Please also provide contact information for a data access committee, ethics committee, or other institutional body to which data requests may be sent.

d. If these are not third party data and there are no restrictions, please upload the minimal anonymized dataset necessary to replicate your study findings to a stable, public repository and provide us with the relevant URLs, DOIs, or accession numbers. For a list of recommended repositories, please see https://journals.plos.org/plosone/s/recommended-repositories. You also have the option of uploading the data as Supporting Information files, but we would recommend depositing data directly to a data repository if possible.

The data on which the article is based are not third-party data. They were collected in each country by the authors. There are no ethical or legal restrictions on sharing the data set publicly, since it is fully anonymous and presents minimal risk to the confidentiality of study participants. The database with the data set necessary to replicate our study is publicly available in the repository located on the project website. We have updated our Data Availability statement to reflect this information and provided more information on the location from which the data set can be downloaded.

3. Please confirm there will be persistent and/or long-term data storage and availability in the database link you've provided.

We confirm that the database with the dataset necessary to replicate the study will be permanently and long-term stored in the repository located on the project website and publicly available at the link provided (http://www.equity-la.eu/en/publicaciones.php?t=IS).

Reviewer 1

1. I thank the authors for their thoughtful consideration of my prior comments, many of my prior concerns were addressed in the revised manuscript. My only remining concern is the substantial discussion of implementation factors in the discussion section that came from previously reported qualitative data. I fully understand that these previously published qualitative data help provide context and insight regarding the quantitative results reported in this paper. However, parts of the current discussion section (e.g., pg 25, line 460 through pg 27, line522) appears to largely re-state findings previously described in Vargas et al., (2020, BMC HSR). Rather than restate these qualitative findings verbatim, I encourage the authors to condense this section and point readers to the prior paper. 

Once again, we would like to thank the reviewer for all their comments, which have helped us to improve the manuscript. In relation to the concern expressed regarding the use of data from the qualitative evaluation in the interpretation of the effectiveness results, we have made an even greater effort here to synthetize them (line 588 to 665). In doing so, we would also like to highlight, firstly, that we have checked that the discussion section does not contain qualitative findings verbatim but synthesizes the evidence found in the two papers already published [5,6], and in another one underway [7]; and secondly, in this section (line 588 to 633), we have checked the relevance of the other international and national studies we used to support the discussion e.g. to interpret the worse results of the intervention network in Colombia (line 592-617), differences between effectiveness and process outcomes (line 622-23), or time as a determinant for the effectiveness of interventions (line 627-633).

Reviewer 3

1. Thank you for the opportunity to review this manuscript that describes one piece of a very ambitious project. The aim of this study was to evaluate the effectiveness of PAR interventions in improving clinical coordination between care levels. This paper has already been through one round of reviews and I am joining as a reviewer in the second round. Unfortunately, I must highlight issues that really need to be addressed, that were not highlighted in the first round.

First, I want to congratulate the whole team for this large project that spanned five (actually, six) countries using a challenging, complex intervention (PAR) with complex methods. There is a LOT to cover in this paper and the authors have clearly improved the paper from the first round. However, key issues must be addressed. These suggestions will help make this paper much clearer and should help strengthen the framing of results by being more transparent about methods and rationale. One note: it was difficult for me to track responses to prior reviews because line and page numbers did not match the content described in each response.

MAJOR REVISIONS

1. This is an easy fix but a major problem: the authors use the term “intervention” throughout. At times “intervention” seems to refer to the PAR approach used and other times, it seems to refer to the “solution” chosen by each country. If my understanding is accurate, I suggest this terminology:

a. Identified gap/problem: this is the “problematized” issue identified by baseline data(?) [line 165] within each IN/country

b. Chosen solution: this is the “selected intervention(s)” [line 165] within the INs

c. PAR Approach: three general approaches were used by the five countries [lines 166-170] – but each country followed very different PAR approaches that don’t seem to align with the 3 types (ref Table S1).

We would like first to thank the reviewer for the positive comments, the interest in our paper, and the suggestions, which have helped us to clarify and improve important aspects of the manuscript related to the description of the interventions, the participatory process and the methodology applied in the study.

With regard to the terms used, we agree on the importance of distinguishing the selected problems and chosen interventions from the PAR process of designing and implementing the interventions. 

In each country, following the dissemination and discussion of the baseline results with the health professionals in the intervention network, they selected in separate and subsequent phases, first, the care coordination problem(s) to be addressed, and then, the interventions to be implemented in order to address them (page 7, line 183). Thus the interventions were the care coordination strategies selected, designed and implemented by the healthcare professionals in each intervention network (line 197-207, and Table 1. Characteristics of the interventions and type of PAR process developed for the selection, design and implementation of the interventions), and the Participatory Action Research process, the participatory process followed for the selection of problems/interventions, design and implementation of the interventions (pp. 8, line 185-190, and Table 1). 

The main problems chosen in all countries were the lack of communication and agreement over patient clinical management between PC and SC doctors, whereas the interventions chosen were: joint meetings of PC and SC doctors to discuss clinical cases and/or for training (Brazil, Chile and Colombia), shared clinical guidelines (Argentina and Brazil); a multi-component strategy to promote the use of the referral and back-referral form (Uruguay); offline virtual consultations between PC and SC doctors (Brazil and Mexico) and an induction program to promote a common identity and shared vision (Chile). 

The process of selecting problems and interventions and of designing, planning and implementing the selected interventions took place through PAR processes. Although these processes differed across countries, in terms of duration, number of PAR cycles and levels of participation[7], three types of participatory processes can be distinguished: 1) with two PAR cycles (Brazil, Mexico and Colombia); 2) a longer, more reflective, flexible PAR process in Chile, including pilot testing of the components of the interventions; and 3) an open design with elements to be specified in Argentina and Uruguay. 

We realize that the utilization of the term “PAR interventions” throughout the manuscript to refer to interventions implemented to improve clinical coordination (joint meetings, offline virtual consultations, etc….), by virtue of mentioning the process, becomes ambiguous and generates confusion. To avoid this, following the reviewer’s suggestion, we have replaced it with “care coordination interventions designed and implemented through a PAR approach” or, where a shorter expression is required, “interventions based on/developed though PAR processes”. 

We have also carefully revised the text to ensure that the distinction between care coordination problems, selected interventions and the PAR process by which they were designed and implemented remains clear throughout the text (pp. 7, line 182-190; pp. 8, 196-207).

2. Note, throughout, five countries are referenced but in Line 185, six countries are referenced. At Line 170, it states that Argentina was dropped because the “selected intervention” was not implemented. The authors need to provide better justification of dropping this country. Using an “intent to treat” approach, Argentina should be included. However, the authors could frame this analysis as including only “completers” or “active participants.” Either way, the failure of the Argentinian IN to implement their chosen intervention (aka chosen solution) needs to be discussed in limitations. Previously published findings all seem to have included all six countries.

The intervention selected in Argentina was shared care guidelines for hypertension and diabetes. However, due to difficulties related to the context (interference of the political cycle) and particularly the research team (turnover of team members), the interventions were selected and planned in general terms, but were never specifically programmed or implemented. At the time of evaluation, the plans had not yet been finalized, hence the dissemination of guidelines to healthcare professionals, and training in their use, were not carried out. Since the guidelines had not been introduced into the network, nor adopted, changes in the network’s coordination attributable to the intervention could not be expected. For this reason, Argentina was excluded from the analysis in this article, as it also was from previous publications describing the qualitative evaluation performed with participants on perceived intervention outcomes and factors that influenced their implementation [5,6]. The analysis of the design process of the intervention and factors that hindered it in Argentina, and the other participating countries, is covered in an ongoing publication [7]. 

We would like to highlight that this is a quasi-experimental study, in which we are not analysing the effectiveness of the interventions across countries, but in each country separately, and the results are interpreted according to the process and factors that are influential in each context, in order to learn from the different experiences. This is the reason why we did not include Argentina in the analysis conducted in this paper.

We have now included more information to help readers understand the factors that hindered the implementation of the intervention in Argentina, as well as its exclusion from the analysis (page 8, line 190-195).

3. In response to the first item from Reviewer #2, the authors present a nice summary of previously published findings that maps out the work already done and how it informs the current manuscript. I suggest reworking the Introduction to better walk readers through what already has been reported and what this manuscript adds. The basic story arc seems to be this: you’ve reported on the PAR approaches, the “problems/gaps” and “solutions” chosen by each IN within each country and now, your aim is “to evaluate the effectiveness of PAR interventions in improving clinical coordination between care levels.”

Following the reviewer’s recommendation, we have provided further information on the previously conducted qualitative evaluation of the implementation of the interventions and on what this manuscript adds (pp. 7, line 162-169).

4. It would be more accurate to state that your aim is “to evaluate the effectiveness of PAR approaches on indicators (or perceptions) of clinical coordination between care levels.” � you are reporting perceptions/self-reported measures via the COORDENA survey instrument, not actual measures of “coordinated care.”

We thank the reviewer for the comment and suggestion. We would like to point out that our study is oriented by a theoretical framework that defines clinical coordination between levels of care as a complex and multidimensional phenomenon, comprising different types - information coordination and clinical management coordination - each with different dimensions, and also organizational and individual factors that may have different influences according to the type and dimension of care coordination [3–5]. This framework also implies a comprehensive analysis, which thus requires the application of comprehensive instruments. From the perspective of the health services, the analysis can be conducted based on the experience and opinions of the professionals involved, by means of surveys or qualitative studies, and/or - based on a review of records, using health services indicators. In this study we used the COORDENA survey, one of the few instruments available for the comprehensive measurement of clinical coordination as experienced by PC and SC doctors [4,8], and already applied in different countries in LA and Europe by this team and others. It could have been complemented by an analysis of indicators, but unfortunately, as we already mentioned in the limitations of the paper, these were not available in the study networks. 

We have revised the description of the theoretical framework and study to make sure that our approach is understood (line 128-151). We have also revised the wording of the objective to make sure that it is clearly stated - to evaluate the effectiveness of the interventions in relation to their stated objective, i.e. the improvement of clinical coordination between levels of care (line 233-236). 

5. Consider moving Table S1 to the main manuscript. By restructuring this table, it can quickly “tell the story” of each country. Here’s a suggested layout:

i. Title � Table 1: Description of Problems, Solutions, and PAR Approaches used by Country

ii. The current Table S1 has this footnote: “More details on the characteristics of each intervention in: http://www.equity-la.eu/en/publicaciones.php?t=PR” but this link goes to presentations without a description of “interventions” (and consider noting that content is in Spanish)

iii. Rows:

1. Problem(s) (e.g., lack of coordination for diabetes patients)

2. Solution(s) (e.g., create and implement guidelines shared by PC and SC providers that focus on essential practices and care pathways)

3. PAR Approach – Joint Meetings:

a. Description (e.g., Discuss mental health care cases face-to-face

b. Who (e.g., PC teams and psychiatrists)

c. Dose (e.g., 11 design(?) sessions with PC and SC providers)

d. Duration (e.g., 7 months)

4. PAR Approach – Virtual Consultations

a. Description (e.g., Asynchronous virtual consultations in Mental Health via email)

b. Who (e.g., PC doctors and psychiatrists)

c. Dose (e.g., 11 consultations)

d. Duration (e.g., 3 months)

5. PAR approach – Other

a. Description

b. Who

c. Dose

d. Duration

iv. Columns: list the five countries

We fully realize that the description of the interventions needs to be more accurate. In our answer to Reviewer 2’s first comment above, we have already clarified the difference between the coordination problems selected, the selected interventions, and the process for the design and implementation of the interventions. 

As recommended by the reviewer, we have now moved Table S1 to the main manuscript (line 213) and added a column with the type of PAR process developed in each country. As mentioned above, a detailed analysis of the process of problem selection, intervention design and intervention content is the focus of an article that is close to being finalized and submitted. We would therefore prefer to limit the information provided here to that contained in the table and inside the manuscript. 

In addition, we have replaced the link in the footnote to the table with the link to the project's publication list, which will contain the article on the design and content of the interventions once it has been published.

6. The text description of the 3 types of PAR used (in Lines 166-170) is confusing. Please clarify how these three general approaches align with the content of Table S1/new Table 1 outlined above.

As outlined above, the three types of PAR process refer to the participatory process that was carried out for designing and implementing the selected interventions listed in Table 1. Although PAR processes differed across countries, three types of participatory processes can be distinguished in terms of duration, number of PAR cycles and levels of participation. As mentioned before, we have replaced the term PAR intervention with “care coordination interventions designed and implemented through a PAR approach” or, where a shorter expression is required, “interventions based on/developed through PAR processes”, to avoid misunderstandings. We have also carefully revised the text to ensure that the distinction between the interventions and the PAR process by which they were selected, designed and implemented is clear (pp. 7, line 182-90; pp. 8, 196-207). 

7. The structure and content of all of the Tables needs to be reworked.

a. Table 1 lists “demographic” descriptions of networks and providers within each of the five participating countries. This table is very similar to Table S2 and yet the numbers are different. This needs to be resolved.

Table 1 (now Table 2) and S2 (now S1) show the demographic and employment characteristics of the sample of doctors in the intervention and control network of each country, for both surveys. The first one is the sample of the evaluation survey (2017), and the second one, that of the baseline survey (2015). In order to be more specific, we have renamed the tables indicating the characteristics shown and the year. In addition, we have added the corresponding title in each of the supplementary tables.

8. Table 1 is labeled as “Evaluation Survey (2017)” and yet the footnote in Table 1, for “***” and “**” says that these symbols indicate differences at baseline. These are for differences between IN and CN…at baseline? This Table should focus only on baseline differences, not a mix of both time points.

The footnotes refer to the statistical differences between the characteristics of the 2017 sample of doctors (evaluation) and those of the 2015 sample (baseline survey) for each healthcare network. Although the samples are quite similar, some differences between years were detected for some of the networks. The statistical models built to analyse the changes in outcomes are adjusted for some of these variables to control for possible changes in outcomes unrelated to the interventions.

9. Table 1 was suggested by a reviewer but there is no rationale for the choice of data to show in this table – e.g., sex and age of providers does not seem to have any relationship to the aim. I suggest moving this table to the Supplement except for a few of the rows:

i. From Table 1, toward the bottom: I suggest listing “Enough time during consultation…” and “Knowledge of the Intervention” and “use of/participation in the intervention” be moved to a separate Table to present in Results. These lines are inferred in the Discussion when explaining possible reasons for differences and lack of differences in the care coordination indicators.

As mentioned above, the information shown in Table 1 provides the main demographic, employment and organisational characteristics of the evaluation survey sample. The description of these characteristics (page 14, line 343-366), and their presentation in Table 1, has several purposes: 1) to understand the differences between networks and countries with respect to individual and organisational variables not related to the interventions that may influence coordination between levels of care in the networks; 2) to facilitate the interpretation of some findings, such as the few changes observed in coordination after the intervention in Colombia, attributable to the low participation of specialists due to their working conditions (page 27, line 643-651); 3) to show the values of some variables such as sex, age or level of care that were used as adjustment variables in the Poisson regression models. 

Following the reviewer's recommendations, we have revised the variables included in the table, and eliminated those that have not been used for the above-mentioned purposes, specifically those related to doctors' attitudes towards their jobs (“planning to change job in the next 6 months”; “satisfaction with job”; and “satisfaction with salary”).

Finally, as the reviewer points out, this table also includes some variables related to the penetration of the interventions in the healthcare network: the knowledge and use/participation of the doctors interviewed in the intervention. Penetration is one of the process outcomes of the intervention implementation [9], and provides important information for understanding the effectiveness of interventions: those with greater penetration would be expected to have a higher impact, in this case, on care coordination between levels of care at the network level. We have included these variables in Table 1 because it is a factor that helps to interpret the results obtained in relation to the care coordination outcomes shown in Tables 2-5.

10. Tables 2 and 3 are very confusing. Methods need to better describe what “PR” is and why that was chosen as your main outcome. AND provide rationale for showing both IN versus CN differences in Table 2 PLUS showing PRs for reversed order in time (2015 vs 2017). I’m seeing 100 paired comparisons across these two tables! With no adjustment for repeated comparisons, statistically. See below for more on Methods. This paper should be reviewed by a statistician and/or better framed to avoid reaching conclusions that are beyond what the data can support

Firstly, PR, the acronym of Prevalence Ratio, is defined as prevalence in the exposed population divided by prevalence in the non-exposed population. It is a measure of association recommended for use in cases where the dependent variable is dichotomous. Prevalence is obtained in the descriptive analysis[4], and it is not possible to establish a causal relationship between the exposure and outcome because they are measured simultaneously, or not enough time has passed between one and the other [10]. Under these circumstances, estimations resulting from other models such as odds ratios (OR) may have introduced an overestimation of the changes in the outcomes, especially when the prevalence ratio (PR) is above 10% [11]. Moreover, PRs are highly recommended because they are easier to interpret [12,13].

In our study, to identify changes in the intermediate outcomes (interactional and organizational factors) and distal outcomes (experiences and general perception of clinical care coordination) in 2017 with respect to 2015 for each network, Poisson regression models were estimated with robust variance, using the prevalence ratio (PR) at the 95% confidence interval (CI 95%). Poisson regression models were also estimated to compare the prevalence of intermediate and distal outcomes of the IN with respect to the CN for each year. These models were estimated for each of the outcomes, by network and year, in each study country.

We have added the definition of the acronym PR as a footnote to the tables, and further explained how it was estimated in the methods section (pp. 13, line 319-324).

Secondly, in response to the reviewer´s comment, to simplify the table we have excluded the changes in the outcomes of the control network between years from Table 2 (and therefore also from Table 4) (now Tables 3 and 5), and included these results in a table as supplementary material (Table S2 and S4). In the manuscript, the changes in outcomes between years are described only for the intervention network, and we did indeed consider this option during the drafting of the manuscript. 

We have also changed the year order in the column headings of Tables 2 and 4 (now tables 3 and 5), this was a mistake we had overlooked. Changes in outcomes were estimated for 2017 relative to 2015, and are described as such in the manuscript.

Finally, in relation to the use of a statistical correction test for multiple comparisons, we would first like to highlight that the analyses were carried out in the networks of each country (no statistical comparisons are made across countries), and 13 outcomes per network were compared in total.

After review by two external statisticians, it was considered more appropriate not to run a correction test for multiple comparisons for several reasons:

1) Bonferroni corrections (and other similar corrections) have been widely criticized. A detailed justification of these criticisms is presented in a paper by Perneger et al [1]. The main weaknesses are that the interpretation of findings depends on the number of tests performed and that Bonferroni corrections increase the probability of type II errors. It is therefore recommended to only make pre-planned and clinically justifiable contrasts of hypothesis, and to clearly describe all the analyses performed so that the reader is aware of the risk of statistically significant differences being observed by chance. 

2) In this regard, our analyses were pre-planned [2] and based on theoretical assumptions [3–5]. This is a quasi-experimental (controlled before-and-after) design, which analysed the effectiveness of interventions in relation to their expected outcomes in improving coordination between levels of care. The selected interventions are multifaceted, which means that they address not only the selected problem, i.e. lack of communication or clinical agreement, but also – on being based on direct feedback between professionals – interactional factors that influence coordination such as mistrust, not knowing each other in person, identifying PC doctors as the coordinators of patient care across care levels, etc. For this reason, our approach, based on the expected outcomes of the interventions, was to select those items from the COORDENA questionnaire that were susceptible to being influenced by the interventions (study hypothesis).

3) Care coordination is a multidimensional phenomenon involving different types and dimensions, as well as many factors that may influence it in different ways. This means that there is no main outcome or hierarchy, thus the items were analysed separately, on an individual basis, since they all address different aspects of care coordination targeted by the intervention.

4) The items used for measuring intermediate and distal outcomes do not constitute (sub)scales, since each item measures a different aspect of clinical care coordination and hence provides distinct information. 

Following the above recommendations, we have carefully revised the methods section. In particular, we have improved the description and justification of the analysis carried out, and added the study's hypotheses on the expected outcomes (page 12, line 292-297 and page 13, line 318). We have also reviewed the limitations of the study, to ensure that the indication regarding the need to treat the results with caution due to the limited number of networks studied in each country has been included (page 30, line 713-16). 

11. The Methods Section needs to be significantly strengthened.

a. Start with the description of your care coordination framework.

i. Readers need more information on the components of care coordination as conceptualized within the COORDENA questionnaire. Readers are not familiar with COORDENA. It seems that CC (care coordination) comprises 1) interactional factors, 2) Organizational Factors; 3) Coordination of information; 4) Consistency of care ; and 5) patient follow-up

1. You conceptualize 1-2 as “intermediate” and 3-5 as “distal.” More accurate labels might be determinants of CC versus perceptions of CC…or other similar labeling

ii. There is an implied causal/influential pathway: “Enough time during consultation…,” “Knowledge of the PAR intervention,” and “Use of/participation in PAR” (listed in Table 1) may influence Interactional and Organizational factors (listed in Tables 2-3) which may lead to (influence) Coordination of Information, Consistency of care, and Patient follow-up (listed in Tables 4-5). Please describe your hypothesized pathway and intent to explore associations along this pathway.

The comprehensive conceptual framework on clinical coordination between levels of care on which this study is based is now described extensively at the end of the introduction (page 5, line 126-151), as Reviewer 1 asked us to move it from the methods section to the introduction. We have included a sentence in the methods section to remind the reader that the framework orientating the study is described in the introduction (page 12, line 295-297). 

In relation to the COORDENA questionnaire, we would like to point out that it was developed based on the conceptual framework and is divided into several sections. The first section contains: a) 12 items to measure clinical care coordination across levels of care experienced by doctors related to i) clinical information coordination: three items on transfer of information; ii) clinical management coordination: four items on care coherence, four items on the follow-up of patients, and two items on accessibility; and b) one item on general perception of care coordination. This is followed by a section on doctors’ interactional factors. To measure distal and intermediate outcomes related to the objectives of the interventions, several items from both of these sections were selected based on their hypothesized impact. The third and fourth sections of the questionnaire refer to the knowledge and use of care coordination mechanisms. The questionnaire applied in 2017 also included a specific subsection to analyse the knowledge and use of the implemented intervention. This section is followed by one on suggestions for improving coordination between levels of care. The penultimate section refers to organizational and employment factors and job-related attitudes, and the final one to demographic characteristics. The variables displayed in Table 1 are drawn from the latter two sections. 

We have revised and improved the description of the structure of the COORDENA questionnaire, to make it clearer how it relates to the conceptual framework for coordination (page 11, line 281-289). The design of the questionnaire and its structure is also described in detail in a previous publication [4].

The terminology for referring to outcomes as distal and intermediate was suggested by Reviewer 1. We have carefully checked the manuscript and tables to ensure that this terminology also includes the variables to which it refers: distal - experience of clinical management coordination and general perception of coordination between levels; and intermediate - interactional and organizational factors.

Finally, in relation to exploring a causal pathway and possible associations, we would like to clarify that the aim of the study is to analyse the effectiveness of the interventions in improving clinical coordination between levels, i.e. to measure the changes in care coordination outcomes between years and networks. Quantitative analysis of the determinants of changes in care coordination outcomes, although interesting and not discarded for future publications, is beyond the scope of this article. In order to analyse the determinants of changes between years, it would be necessary to generate explanatory models for each of the outcomes and the study network, and explore other variables that may have influenced the changes, such as the use of other care coordination mechanisms, acceptability of the interventions, etc. Reporting all these results would require a separate article. 

12. Study Design

You characterize this study as “quasi-experimental” which not a helpful label. It would be more accurate to describe this as a prospective non-randomized trial. You pre-selected INs and CNs and then tested the effects of PAR on perceptions of CC (care coordination). Each network identified different problems/gaps and solution(s) and each used a different mix of PAR approaches

In this study, we adopt the concept of "quasi-experimental design" proposed by Campbell [14], as opposed to experimental designs. It refers to those designs in which, for reasons of context, practicality, etc., the researchers cannot conduct a genuine (randomised) experiment, and have little control over the delivery of an intervention [15]. We consider the term quasi-experimental to be appropriate to describe the design of this study for several reasons: 1) it is an evaluation of the effectiveness of clinical coordination interventions designed and implemented through a PAR process, in which – due to the participatory and collaborative nature of the research – the researcher does not control many aspects, such as the allocation of the intervention and control network, the problems and type of intervention to be implemented, comparability between the IN and CN, etc.; 2) in the field of community-based evaluation, where the PAR approach has been most widely used and evaluated, it is the term most commonly used to refer to this type of design [16–18]; 3) quasi-experimental is also the term we have used in previous publications to describe the study design [3,5,6], and we prefer to continue using it for the sake of consistency.

We have added the bibliographic reference to Campbell’s papers in the definition of the study design (page 10, line 239) 

13. Study Population and Sample

Your unit of analysis seems to be “Country (or Network(?))” – an intervention network (IN) and control network (CN), which are each in separate countries. State this. 

The unit of analysis is the healthcare network. In each study country, two comparable healthcare networks were selected – an intervention network (IN) and a control network (CN). 

We have rewritten the study population and sample section to clarify these issues (page 10, line 245-246), and described the unit of analysis (page 13, line 314) 

14. Within each country/network, you sampled doctors at two levels: 1) primary; and 2) secondary. Define what types of care are included in each of these levels. These providers are clustered by network – this forms the basis for your outcomes.

As described in the introduction (page 10, line 223), the study countries have different health system models. However, healthcare provision in the public subsystem, which is the focus of this study, is organized by levels of complexity, with primary care (PC) as the entry point and coordinator of patient care and secondary/tertiary care (SC) in a supporting role, requiring a referral from PC for access to the specialist. SC includes different types of health care: outpatient secondary care, emergency care and inpatient care. We have added this information in the introduction of the manuscript (page 10, line 227-232). 

15. Primary outcome

i.Hundreds of paired comparisons are presented. Any statistician (and I am not a statistician) will warn against multiple comparisons – and any conclusions based on a seeming random assortment of “significance” from among the hundreds.

ii. A single primary analysis is needed. The detailed paired comparisons can be presented (in a more limited and thoughtful way) to help explain (negative?) results of your primary analysis.

iii. Consider a table that shows baseline results for IN and CNs overall, with ranges across the countries.

Our answer to the comments regarding the multiple comparisons between outcomes and countries, and the type of analysis that was carried out, is given in point 10.

15. What is “PR” – how is it computed? This seems to be your primary outcome. However, it is not used appropriately. A single set of ratios – or difference in difference scores? – should be listed for each of the measures

As mentioned above (point 10), PR is the prevalence ratio. In our study, Poisson regression models with robust variance were estimated to test the hypotheses on the possible influence of the intervention on intermediate and distal outcomes. Prevalence ratios (PR) and respective 95% confidence intervals (CI 95%) were calculated in each country, first, to compare changes in the outcomes for each network (IN and CN) after the intervention; and second, to compare the prevalence of the outcome variables of the IN with respect to CN both at baseline and post-intervention. 

We have added it as footnote to the tables and its computation is explained in the methods section. We have also revised captions to ensure that all relevant information is provided. 

We agree with the reviewer on simplifying the tables and listing a single set of ratios for each measure, in line with the analysis carried out. Tables 2 and 4 (now Table 3 and 5) now only show the set of ratios for the intervention network by year and, as mentioned above, the results for the control network have been moved to Supplementary Table S2 and S4. Tables 3 and 5 (now Table 4 and 6) show the PRs from the comparison between networks after the intervention (2017), and those from 2015 have been included in Supplementary Table S3 and S5. This is even more consistent with the type of analysis conducted, as primarily the difference between networks after the intervention (2017) was analysed, and only if there were any significant differences (very few were observed) was the difference between networks at baseline analysed (2015). With regard to the application of another analytical approach, such as the difference in difference method, it would not be appropriate for this study because the variables are non-continuous and it is not possible to follow the subjects in the groups[19,20].

16. Regarding measurement: will the COORDENA measures support “scale-level” averages for each of the 5 dimensions (e.g., organizational factors as a single averaged score, instead of comparing every item (e.g., a single indicator for organizational factors instead of 2 separate items))?

Regarding the sum scales in the COORDENA questionnaire, the only two sections that could theoretically represent sub-scales are the first section, which includes items related to experience of clinical coordination, and the second section on doctors’ interactional factors, as both sections use a Likert scale (always, often, sometimes, rarely, never). However, as highlighted in point 10, these items in the COORDENA questionnaire do not constitute (sub)scales, since each item measures a different aspect of clinical care coordination and hence provides distinct information, so they are not correlated. They are therefore assessed separately, on an individual basis. 

In any case, this would not be appropriate for the purposes of the study because it was not the whole set of items from section 1, or of interactional factors from section 2, that were selected, but rather only those items that were susceptible to being influenced by the interventions (study hypothesis).

17. When the measures are better defined as suggested above, along with an appropriate analytic plan, write up results that follow that flow: present primary outcomes followed by exploratory analyses to help explain findings.

Following the suggestions of the Editor and Reviewer 2, and as described in previous answers, we have revised the whole manuscript in order to help the reader clearly understand and follow the main message from the study findings. 

As stated in point 11, an explanatory analysis of the determinants of changes in coordination between levels of care following the intervention in each network is beyond the scope of this article.

18. Table S6 is a nice summary of “significant” findings but combining “significance” for IN vs CN comparison with baseline versus follow-up within IN-only, into the same table, is not theoretically justified.

Following the reviewer’s recommendation, we have simplified the table S6 (now table S9) and focused on improvement in INs between 2015 and 2017, to be consistent with the methods description. 

References

1. Perneger T V. What’s wrong with Bonferroni adjustments. BMJ. 1998;316: 1236–1238. doi:10.1136/bmj.316.7139.1236

2. Equity-LA II. Cross-country comparative analytical plan to analyze the effectiveness of PAR interventions to improve clinical coordination across care levels in healthcare networks of Brazil, Chile, Colombia, Mexico and Uruguay. Barcelona; 2018. 

3. Vázquez ML, Vargas I, Unger JP, De Paepe P, Mogollón-Pérez AS, Samico I, et al. Evaluating the effectiveness of care integration strategies in different healthcare systems in Latin America: the EQUITY-LA II quasi-experimental study protocol. BMJ Open. 2015;5: e007037. doi:10.1136/bmjopen-2014-007037

4. Vazquez M, Vargas I, Garcia-Subirats I, Unger J-PP, de Paepe P, Mogollón-Pérez AS, et al. Doctors’ experience of coordination across care levels and associated factors. A cross-sectional study in public healthcare networks of six Latin American countries. Soc Sci Med. 2017;182: 10–19. doi:10.1016/j.socscimed.2017.04.001

5. Vargas I, Eguiguren P, Mogollón-Pérez A-S, Samico I, Bertolotto F, López-Vázquez J, et al. Can care coordination across levels be improved through the implementation of participatory action research interventions? Outcomes and conditions for sustaining changes in five Latin American countries. BMC Health Serv Res. 2020;20: 941. doi:10.1186/s12913-020-05781-7

6. Vargas I, Vázquez M-LL, Eguiguren P, Mogollón-Pérez A-S, Bertolotto F, Samico I, et al. Understanding the factors influencing the implementation of participatory interventions to improve care coordination. An analytical framework based on an evaluation in Latin America. Health Policy Plan. 2020;35: 962–972. doi:10.1093/heapol/czaa066

7. Vargas I, Eguiguren P, Mogollón A, López J, Samico I, Bertolotto F, et al. [Participatory design of interventions to improve clinical coordination in Latin America]. Gaceta Sanitaria. 2019;33:SC. 

8. Esteve-Matalí L, Vargas I, Sánchez E, Ramon I, Plaja P, Vázquez M. Do primary and secondary care doctors have a different experience and perception of cross-level clinical coordination? Results of a cross-sectional study in the Catalan National Health System (Spain). BMC Fam Pract. 2020;21. doi:10.1186/s12875-020-01207-9

9. Proctor E, Silmere H, Raghavan R, Hovmand P, Aarons G, Bunger A, et al. Outcomes for implementation research: conceptual distinctions, measurement challenges, and research agenda. Adm Policy Ment Heal. 2011;38: 65–76. Available: http://www.ncbi.nlm.nih.gov/pubmed/20957426

10. Carlson MDA, Morrison RS. Study design, precision, and validity in observational studies. J Palliat Med. 2009;12: 77–82. doi:10.1089/jpm.2008.9690

11. Szklo M, Nieto FJ. Epidemiology: Beyond the basics. 3rd ed. Jones & Bartlett Learning; 2012. 

12. Grimes DA, Schulz KF. Making sense of odds and odds ratios. Obstet Gynecol. 2008;111: 423–426. doi:10.1097/01.AOG.0000297304.32187.5d

13. Zocchetti C, Consonni D, Bertazzi PA. Relationship between prevalence rate ratios and odds ratios in cross-sectional studies. Int J Epidemiol. 1997;26: 220–223. doi:10.1093/ije/26.1.220

14. Campbell DT, Stanley JC. Experimental and quasi-experimental designs for research. Houghton Mifflin Company; 1963. 

15. Eccles M, Grimshaw J, Campbell M, Ramsay C. Research designs for studies evaluating the effectiveness of change and improvement strategies. Quality and Safety in Health Care BMJ Publishing Group; Feb, 2003 pp. 47–52. doi:10.1136/qhc.12.1.47

16. Viswanathan M, Ammerman A, Eng E, Garlehner G, Lohr KN, Griffith D, et al. Community-based participatory research: assessing the evidence. Evid Rep Technol Assess. 2004; 1–8. Available: https://www.ncbi.nlm.nih.gov/books/NBK11852/

17. Merzel C, D’Afflitti J. Reconsidering community-based health promotion: Promise, performance, and potential. Am J Public Health. 2003;93: 557–574. doi:10.2105/AJPH.93.4.557

18. Nickel S, von dem Knesebeck O. Effectiveness of Community-Based Health Promotion Interventions in Urban Areas: A Systematic Review. J Community Health. 2020;45: 419–434. doi:10.1007/s10900-019-00733-7

19. Fredriksson A, Oliveira GM de. Impact evaluation using Difference-in-Differences. RAUSP Manag J. 2019;54: 521. doi:10.1108/RAUSP-05-2019-0112

20. Lechner M. The estimation of causal effects by difference-in-difference methods. Found Trends Econom. 2010;4: 168. doi:10.1561/0800000014

---

## [Decision Letter · Decision Letter 2]

7 Dec 2021

Evaluating the effectiveness of care coordination interventions designed and implemented through a participatory action research process: lessons learned from a quasi-experimental study in public healthcare networks in Latin America

PONE-D-20-36859R2

Dear Dr. Vargas,

We’re pleased to inform you that your manuscript has been judged scientifically suitable for publication and will be formally accepted for publication once it meets all outstanding technical requirements.

Kind regards,

Sylvia J. Hysong

Guest Editor

PLOS ONE

Reviewers' comments:

Reviewer's Responses to Questions

**Comments to the Author**

1. If the authors have adequately addressed your comments raised in a previous round of review and you feel that this manuscript is now acceptable for publication, you may indicate that here to bypass the “Comments to the Author” section, enter your conflict of interest statement in the “Confidential to Editor” section, and submit your "Accept" recommendation.

Reviewer #1: All comments have been addressed

Reviewer #4: All comments have been addressed

2. Is the manuscript technically sound, and do the data support the conclusions?

Reviewer #1: Yes

Reviewer #4: Partly

3. Has the statistical analysis been performed appropriately and rigorously? 

Reviewer #1: Yes

Reviewer #4: I Don't Know

4. Have the authors made all data underlying the findings in their manuscript fully available?

Reviewer #1: Yes

Reviewer #4: Yes

5. Is the manuscript presented in an intelligible fashion and written in standard English?

Reviewer #1: Yes

Reviewer #4: Yes

6. Review Comments to the Author

Reviewer #1: I appreciate the thoroughness and thoughtfulness of the authors’ reply and revisions. This revision is responsive to key feedback to clarify presentation of study findings and more clearly orient this set of findings in the context of the larger study. I do offer two very minor comments below, however, in my opinion the revised manuscript satisfies the PLOS One criteria for publication.

1. I do think the text of the results section could be condensed—much of the text repeats information in the tables—and the discussion section could be condensed, however, that is a minor stylistic preference.

2. It would be important to note in the introduction or limitations section that PAR approaches involving front line clinicians, staff, and administrators are often used in unpublished quality improvement work and care delivery improvement collaboratives.

Reviewer #4: I read this paper with great anticipation. It represents a very ambitious study and a great deal of work. Unfortunately, I was left with more questions than when I started. That is not to say that the manuscript is not scientifically rigorous. I think that it is, albeit with limitations that are mentioned by the authors. My issues are the following:

(1) have the authors made the case that the outcomes on physician perceptions of coordination can be attributed to a participatory action reaction approach? I think that the case is weak. That there have been some changes in networks that used PAR, there were networks used PAR and saw no changes. This is a case where a follow up qualitative study would be very helpful (but i think that it is much too much to ask of the authors at this point). The changes, such as they are, consist of "statistically significant" differences. Although i agree that using a Bonferroni correction for multiple tests is too conservative, at least the authors should be clear about the limitations because the results are basically all over the place. Are these clinically significant differences anyway? They provide a power analysis, but what is the basis of the choice of a 15% difference? The comparisons are basically between two networks. To draw any conclusions, it is critical that the networks be comparable. This goes beyond numbers, but context. If this has previously been reported and this paper is only a small slice of their study data, there has to be a statement like "These networks (within country) have previously been shown to be comparable base on ..." Parenthetically, publishing all the papers together as a monograph would seem to be a logical approach to address this.

(2) The PAR is not really detailed very well, though it is improved compared to the initial submission. I realize that there are limitations on the length of a manuscript. But there should be enough for me to distinguish PAR from QI from a design science approach. Frankly, i think that only the names have been changed or it is a case that there is more than one way to skin a cat and in Europe we do X while in the US we do Y?

(3) There is much mention of the importance of context, but little description of what the context really is apart from things like appointment length. Do all the networks (besides Argentina) have EHRs?

(4) This a very difficult read, although clearer than the first version. You have to be prepared for a tough slog without the prospect of a big reward.

In the end, I don't object to publishing the paper, but i do think that the authors need to pull back on their conclusions somewhat.

7. PLOS authors have the option to publish the peer review history of their article (what does this mean?). If published, this will include your full peer review and any attached files.

Reviewer #1: No

Reviewer #4: No

---

## [Editor Report · Acceptance letter]

21 Dec 2021

PONE-D-20-36859R2 

Evaluating the effectiveness of care coordination interventions designed and implemented through a participatory action research process: lessons learned from a quasi-experimental study in public healthcare networks in Latin America. 

Dear Dr. Vargas:

I'm pleased to inform you that your manuscript has been deemed suitable for publication in PLOS ONE. Congratulations! Your manuscript is now with our production department. 

Kind regards, 

on behalf of

Dr. Sylvia J. Hysong 

Guest Editor

PLOS ONE